CERN-TH-2025-039

# U(1) Gauging, Continuous TQFTs, and Higher Symmetry Structures

Adrien Arbalestrier,[1] Riccardo Argurio,[1] and Luigi Tizzano[2]

[1] *Physique Théorique et Mathématique and International Solvay Institutes*
*Université Libre de Bruxelles; C.P. 231, 1050 Brussels, Belgium*

[2] *CERN, Theoretical Physics Department, CH-1211 Geneva 23, Switzerland*

Quantum field theories can exhibit various generalized symmetry structures, among which higher-group symmetries and non-invertible symmetry defects are particularly prominent. In this work, we explore a new general scenario in which these two structures are intertwined. This phenomenon arises in four dimensions when gauging one of multiple $U(1)$ 0-form symmetries in the presence of mixed 't Hooft anomalies. We illustrate this with two distinct models that flow to an IR gapless phase and a gapped phase, respectively, and examine how this symmetry structure manifests in each case. Additionally, we investigate a five-dimensional model where a similar structure exists intrinsically. Our main tool is a symmetry TQFT in one higher dimension, formulated using non-compact gauge fields and having infinitely many topological operators. We carefully determine its boundary conditions and provide a detailed discussion on various dressing choices for its bulk topological operators.

February 2025

## 1. Introduction

One of the most commonplace processes in quantum field theory is the dynamical gauging of an ordinary (zero-form) $U(1)^{(0)}$ global symmetry, which involves introducing a new degree of freedom — a photon. Despite its simplicity, in recent years, there has been a renewed understanding of certain physical consequences arising from such gauging in the presence of other global symmetries, which modify the conservation law of the $U(1)^{(0)}$

symmetry current. The two main examples discussed in the literature are:

1) **2-group Global Symmetries** [1–8]: In this case, the theory obtained after dynamically gauging $U(1)^{(0)}$ develops an emergent one-form [9] $U(1)^{(1)}$ global symmetry that mixes with (at least) another zero-form symmetry $G_{\mathsf{A}}^{(0)}$ in the system, leading to unconventional Ward identities that constrain the allowed patterns of spontaneous $G_{\mathsf{A}}^{(0)}$ symmetry breaking [4].

2) **Non-invertible Global Symmetries** [10–16]: Here, the theory arising after dynamical gauging retains an ordinary global symmetry which, however, becomes non-invertible. A primary example of this phenomenon is the axial symmetry in massless QED [17, 18], see [19] for recent updates.[1]

The goal of this work is to discuss a natural extension of both ideas, which we dub *generalized Abelian symmetry structure*. Specifically, we aim to explore what occurs in models where both phenomena 1) and 2) coexist. Our interest in this question stems from the observation that many models with continuous generalized symmetries, across various spacetime dimensions, exhibit both of these unexpected features [27–33].

On a more technical level, we address several related questions. In the original analysis of [4], one of the key assumptions for introducing continuous 2-group global symmetries is precisely the absence of non-conservation laws that would lead to the phenomenon described in point 2) above. At the same time, in a theory with a non-invertible continuous global symmetry, it is unclear how to define background gauge fields for such a symmetry, thus complicating the discussion of the phenomenon in 1).

In order to relax the assumption from [4] and incorporate non-invertible global symmetries, we will adopt the framework of Symmetry TFT [9, 34–40], which was recently extended to discuss $U(1)^{(0)}$ global symmetries in [41–43] (see also [44] for continuous non-Abelian global symmetries). This framework will need to be suitably modified, as we will describe in the main body of this work. Due to its general nature, we believe that this approach will also be valuable for future studies on $U(1)$ gaugings in other contexts.

After introducing in Section 2 the Symmetry TFT framework for generalized Abelian symmetry structures, we will examine a few simple and explicit models that exhibit these features in Section 3. Our analysis will focus on two four-dimensional models, where we will investigate the fate of the symmetry structure under RG flow. First, we will consider

---

[1]Alternative formulations of the non-invertible axial symmetry have been proposed in [20, 21]. In string theory, both $U(1)$ and non-invertible axial symmetry defects have been discussed in [22]. See also [23–26] for further updates.

a model that flows to an IR gapless phase where all the symmetry structure is spontaneously broken. Next, we will examine a model that flows to a gapped phase, where the unbroken symmetry structure is encoded by a topological quantum field theory at low energy. To emphasize that these structures are not limited to four-dimensional physics or the gauging of anomalous symmetries, we will also discuss in Section 4 a toy model in five spacetime dimensions, where the photon's kinetic term is supplemented by a Chern-Simons interaction.

The appendices contain some background material, and some technical complementary details.

**Notation and Conventions**

Throughout this work, depending on the context, for a given object $\mathcal{O}$ the notation $\mathcal{O}^{(p)}$ is used to denote either a $p$-form, a $p$-dimensional manifold or a $p$-dimensional homology cycle. Moreover, we adopt the following convention for denoting $\mathbb{R}$ vs $U(1)$ and background vs dynamical fields:

| font type | lower case | upper case |
|---|---|---|
| serif | $p$-form $\mathbb{R}$ dynamical field : $f^{(p)}$ | $p$-form $U(1)$ dynamical field : $F^{(p)}$ |
| mathsf | $p$-form $\mathbb{R}$ background field: $\mathsf{f}^{(p)}$ | $p$-form $U(1)$ background field: $\mathsf{F}^{(p)}$ |

## 2. Generalized Abelian Symmetry Structures

In this section we present what we believe to be the most general structure in four dimensions involving two continuous Abelian 0-form and 1-form global symmetries. These symmetries mix in a 2-group-like structure; however, the 0-form symmetry is non-invertible. We begin by reviewing how this structure is obtained from a theory with two $U(1)$ 0-form symmetries with mixed 't Hooft anomalies, and dynamically gauging one of them. We then proceed to describe this symmetry structure using a five-dimensional Symmetry TFT, demonstrating how it encodes all aspects of the higher symmetry structure through an appropriate choice of topological boundary conditions. We also clarify that some bulk topological operators need to be made genuine, while others can still be non-genuine but attached to the boundary in a cylinder-like structure.

### 2.1. Abelian Symmetries and their Gaugings in Four Dimensions

The simplest examples of symmetry structures explored in this work can all be constructed from four-dimensional QFTs admitting the following Abelian 0-form global sym-

metries,

$$U(1)_{\mathsf{A}}^{(0)} \times U(1)_{\mathsf{C}}^{(0)} \,, \tag{2.1}$$

where $\mathsf{A}^{(1)}$ and $\mathsf{C}^{(1)}$ are $U(1)$ background gauge fields whose corresponding field strengths are $\mathsf{F}_{\mathsf{A}}^{(2)}$ and $\mathsf{F}_{\mathsf{C}}^{(2)}$. The complete set of 't Hooft anomalies that only involve $U(1)_{\mathsf{A}}^{(0)}$ and $U(1)_{\mathsf{C}}^{(0)}$ is captured by a six-form anomaly polynomial (involving $\mathsf{F}_{\mathsf{A}}^{(2)}$ and $\mathsf{F}_{\mathsf{C}}^{(2)}$) which is reported in Appendix A.

Our interest here lies in obtaining a complete characterization of the symmetry structure that emerges upon *gauging* the symmetry $U(1)_{\mathsf{C}}^{(0)}$ in (2.1). This is obtained by promoting $\mathsf{C}^{(1)}$ and its field strength $\mathsf{F}_{\mathsf{C}}^{(1)}$ to dynamical $U(1)$ gauge fields:

$$\mathsf{C}^{(1)} \to C^{(1)} \,, \quad \mathsf{F}_{\mathsf{C}}^{(1)} \to F_C^{(1)} \,, \tag{2.2}$$

and performing the functional integral over the orbits of $C^{(1)}$. Such gauging procedure is only possible if $k_{\mathsf{C}^3} = 0$, that is, if the cubic $U(1)_{\mathsf{C}}^{(0)}$ 't Hooft anomaly vanishes. Now, the non-conservation equation for the current $j_{\mathsf{A}}^{(1)}$ associated with the symmetry $U(1)_{\mathsf{A}}^{(0)}$ is given by:

$$d * j_{\mathsf{A}}^{(1)} = -\frac{i}{4\pi^2} \left( \frac{k_{\mathsf{A}^3}}{6} \mathsf{F}_{\mathsf{A}}^{(2)} \wedge \mathsf{F}_{\mathsf{A}}^{(2)} + \frac{k_{\mathsf{A}^2\mathsf{C}}}{2} \mathsf{F}_{\mathsf{A}}^{(2)} \wedge \mathsf{F}_{\mathsf{C}}^{(2)} + \frac{k_{\mathsf{A}\mathsf{C}^2}}{2} \mathsf{F}_{\mathsf{C}}^{(2)} \wedge \mathsf{F}_{\mathsf{C}}^{(2)} \right) \,. \tag{2.3}$$

Upon gauging, the background field strength $\mathsf{F}_{\mathsf{C}}^{(1)}$ becomes an operator $F_C^{(1)}$ and therefore the terms proportional to $k_{\mathsf{A}^2\mathsf{C}}$ and $k_{\mathsf{A}\mathsf{C}^2}$ produce operator-valued shifts of the background effective action and can no longer be thought of as 't Hooft anomalies. In the remainder of this section we will describe how to correctly take into account the effects of these terms and how they affect the resulting symmetry structure of the theory.

Let us first analyze the term

$$d * j_{\mathsf{A}}^{(1)} \supset -\frac{ik_{\mathsf{A}^2\mathsf{C}}}{8\pi^2} \mathsf{F}_{\mathsf{A}}^{(2)} \wedge F_C^{(2)} \,. \tag{2.4}$$

Note that, when $\mathsf{F}_{\mathsf{A}}^{(2)} = 0$, the right-hand side of the above equation vanishes and the $U(1)_{\mathsf{A}}^{(0)}$ current $j_{\mathsf{A}}^{(1)}$ is conserved and satisfies $d * j_{\mathsf{A}}^{(1)} = 0$ at separated points. However, the presence of $F_C^{(2)}$ introduces an operator-valued shift of the background effective action which must be cancelled. The modern understanding of the equation (2.4) is that it introduces an Abelian 2-group global symmetry [4]:

$$U(1)_{\mathsf{A}}^{(0)} \times_{-\frac{1}{2}k_{\mathsf{A}^2\mathsf{C}}} U(1)_{\mathsf{C}}^{(1)} \,, \tag{2.5}$$

where $-\frac{1}{2}k_{\mathsf{A}^2\mathsf{C}}$ takes integer values and is typically referred to as 2-group structure constant.[2] The symmetry group $U(1)^{(1)}_{\mathsf{C}}$ is associated with the magnetic 1-form global symmetry with conserved current

$$j^{(2)}_{\mathsf{C}} = \frac{i}{2\pi} * F^{(2)}_C . \tag{2.6}$$

A theory is said to possess an Abelian 2-group global symmetry (2.5) if it is invariant under the combined background gauge field transformations:

$$\mathsf{A}^{(1)} \to \mathsf{A}^{(1)} + d\lambda^{(0)}_{\mathsf{A}} , \qquad \mathsf{C}^{(2)} \to \mathsf{C}^{(2)} + d\lambda^{(1)}_{\mathsf{C}} - \frac{k_{\mathsf{A}^2\mathsf{C}}}{4\pi} d\lambda^{(0)}_{\mathsf{A}} \wedge \mathsf{A}^{(1)} . \tag{2.7}$$

where $\mathsf{C}^{(2)}$ is a $U(1)$ 2-form background gauge field for the $U(1)^{(1)}_{\mathsf{C}}$ symmetry. The basic idea is that the gauge transformation of $\mathsf{C}^{(2)}$ can be modified to require that $\mathsf{C}^{(2)}$ shifts under a $U(1)^{(0)}_{\mathsf{A}}$ background transformation to exactly remove the operator-valued shift induced by (2.4).[3] Abelian continuous 2-groups appear in standard four-dimensional QFTs, as exemplified already in [4, 45–47].

Let us now repeat the previous analysis for the term proportional to $k_{\mathsf{AC}^2}$ in (2.3):

$$d * j^{(1)}_{\mathsf{A}} \supset -\frac{ik_{\mathsf{AC}^2}}{8\pi^2} F^{(2)}_C \wedge F^{(2)}_C . \tag{2.8}$$

This term gives rise to an ABJ anomaly for the $U(1)^{(0)}_{\mathsf{A}}$ current whose conservation is now violated even in absence of background fields. The traditional interpretation of this phenomenon is that the continuous $U(1)^{(0)}_A$ is not a global symmetry of the quantum theory, only the discrete subgroup $\mathbb{Z}_{k_{\mathsf{AC}^2}}$ is preserved. Indeed, one of the basic assumptions in the study of Abelian 2-group symmetries (2.5) is that the symmetry $U(1)^{(0)}_{\mathsf{A}}$ is free of ABJ anomalies, i.e. $k_{\mathsf{AC}^2} = 0$.

The understanding of the ABJ anomaly (2.8) has changed dramatically in recent years. These developments are tied to the interpretation of symmetries in terms of topological defects [9] and the discovery of non-invertible global symmetries in $d > 2$ [15, 16, 48]. Following [17, 18], a symmetry suffering from an anomaly (2.8) can be associated with codimension 1 topological defects $\mathcal{D}_\alpha(\Pi^{(3)})$ implementing $U(1)^{(0)}_A$ rotations of angle $\alpha \in$

---

[2]The general quantization condition on $k_{\mathsf{A}^3}, k_{\mathsf{A}^2\mathsf{C}}, k_{\mathsf{AC}^2}$ is discussed in Appendix A.

[3]By a shift in the definition of $\lambda^{(1)}_{\mathsf{C}}$, the second gauge transformation can also be written as $\mathsf{C}^{(2)} \to \mathsf{C}^{(2)} + d\lambda'^{(1)}_{\mathsf{C}} + \frac{k_{\mathsf{A}^2\mathsf{C}}}{4\pi} \lambda^{(0)}_{\mathsf{A}} F^{(2)}_{\mathsf{A}}$.

$[0, 2\pi)$ which however do not obey the group multiplication law

$$\mathcal{D}_\alpha(\Pi^{(3)}) \times \mathcal{D}_{-\alpha}(\Pi^{(3)}) = \mathcal{C}_{U(1)}(\Pi^{(3)}) \neq 1 \,, \tag{2.9}$$

where $\mathcal{C}_{U(1)}(\Pi^{(3)})$ is a condensation defect [49, 50] for the $U(1)_\mathsf{C}^{(1)}$ magnetic symmetry. To define these operators we follow [19]. The explicit expressions of $\mathcal{D}_\alpha(\Pi^{(3)})$ and $\mathcal{C}_{U(1)}(\Pi^{(3)})$ can be compactly written as follow:

$$\mathcal{D}_\alpha(\Pi^{(3)}) = \exp\left(-\int_{\Pi^{(3)}} \alpha * j_A^{(1)} + \mathcal{T}^{\alpha k_{\mathsf{AC}^2}}[F_C^{(2)}](\Pi^{(3)})\right)\,, \qquad \mathcal{D}_0(\Pi^{(3)}) = \mathcal{C}_{U(1)}(\Pi^{(3)})\,, \tag{2.10}$$

where $\mathcal{T}^{\alpha k_{\mathsf{AC}^2}}[F_C^{(2)}](\Pi^{(3)})$ is a three-dimensional mixed $U(1)/\mathbb{R}$ topological BF theory coupled to the dynamical field strength $F_C^{(2)}$ whose action is

$$\mathcal{T}^{\alpha k_{\mathsf{AC}^2}}[F_C^{(2)}](\Pi^{(3)}) = \frac{i}{2\pi} \int_{\Pi^{(3)}} -\frac{\alpha k_{\mathsf{AC}^2}}{4\pi} c^{(1)} \wedge dc^{(1)} + \Phi^{(1)} \wedge (dc^{(1)} + F_C^{(2)})\,. \tag{2.11}$$

The fields $c^{(1)}$ and $\Phi^{(1)}$ are dynamical connections taking values in $\mathbb{R}$ and in $U(1)$, respectively, which live on $\Pi^{(3)}$ (here and below, we sometimes omit to write explicitly the path integral over the fields localized on the defect). This TQFT can be interpreted as the boundary theory obtained after successively gauging the full $U(1)_\mathsf{C}^{(1)}$ symmetry (with flat connection) and its Pontryagin dual symmetry $\mathbb{Z}_\mathsf{C}^{(1)} = \widehat{U(1)}_\mathsf{C}^{(1)}$ (with torsion proportional to $\alpha k_{\mathsf{AC}^2}$) on a four-dimensional manifold whose boundary is $\Pi^{(3)}$. As a consequence, the operators $\mathcal{D}_\alpha$ act non-invertibly on all line operators charged under the $U(1)_\mathsf{C}^{(1)}$ symmetry. More precisely, when $\mathcal{D}_\alpha$ is moved across a magnetic 't Hooft line operator $H_q$ of $U(1)_\mathsf{C}^{(1)}$ charge $q \in \mathbb{Z}$, the line operator becomes non-genuine:

$$H_q(\gamma^{(1)}) \exp\left(\frac{iq\alpha k_{\mathsf{AC}^2}}{2\pi} \int_{\Sigma_\gamma^{(2)}} F_C^{(2)}\right)\,, \tag{2.12}$$

where $\partial\Sigma_\gamma^{(2)} \subset \gamma^{(1)} \cup \Pi^{(3)}$. On the other hand, $\mathcal{D}_\alpha$ acts invertibly on local operators charged under $U(1)_\mathsf{A}^{(0)}$.

Armed with such a renewed interpretation of Abelian symmetries subject to the ABJ anomaly (2.8), our goal is to examine the general symmetry structure that emerges after gauging (2.2) in a theory where *both* coefficients $k_{\mathsf{A}^2\mathsf{C}}$ and $k_{\mathsf{AC}^2}$ are non-vanishing. To perform this analysis we face a number of technical complications that need to be addressed. First, a symmetry obeying the non-invertible fusion rule (2.9) does not simply

admit a standard coupling to a background gauge field, which takes values in the (invertible) group $U(1)_A^{(0)}$. A related issue is that the term proportional to $k_{A^3}$, which is a 't Hooft anomaly for the $U(1)_A^{(0)}$ symmetry and is unaffected by the gauging (2.2), must now be interpreted differently as the symmetry becomes non-invertible. Finally, unlike the discussion on non-invertible symmetries, most of the literature on (continuous) Abelian 2-group global symmetries, as in (2.5), is presented in terms of background gauge fields rather than in terms of topological symmetry defects and their fusion rules.[4]

## 2.2. Symmetry TFT Analysis

The primary tool used for describing a four-dimensional theory with anomalous $U(1)_A^{(0)} \times U(1)_C^{(0)}$ symmetry is the following five-dimensional Symmetry TFT action:

$$
\begin{aligned}
S_5 = \frac{i}{2\pi} \int_{X^{(5)}} &-b^{(3)} \wedge dA^{(1)} - \widetilde{b}^{(3)} \wedge dC^{(1)} + \frac{k_{AC^2}}{4\pi} A^{(1)} \wedge dC^{(1)} \wedge dC^{(1)} \\
&+ \frac{k_{A^2C}}{4\pi} C^{(1)} \wedge dA^{(1)} \wedge dA^{(1)} + \frac{k_{A^3}}{12\pi} A^{(1)} \wedge dA^{(1)} \wedge dA^{(1)} ,
\end{aligned}
\tag{2.13}
$$

where $b^{(3)}$ and $\widetilde{b}^{(3)}$ are dynamical $\mathbb{R}$ gauge fields while $A^{(1)}$ and $C^{(1)}$ are $U(1)$ dynamical gauge fields. The coefficients $k_{AC^2}, k_{A^2C}, k_{A^3}$ are all integers and $k_{AC^2}, k_{A^2C}$ must have the same parity, see appendix A for further details. This action has been introduced in [41, 42], which we will follow closely.

As discussed in the previous section, if the $U(1)_C^{(0)}$ symmetry is anomaly free it can be dynamically gauged in the four-dimensional physical theory living at the boundary $\partial X^{(5)}$. Gauging the $U(1)_C^{(0)}$ symmetry[5] gives rise to a new symmetry TFT where we perform the substitution:

$$
\begin{cases}
\widetilde{b}^{(3)} \to dC^{(2)} , \\
dC^{(1)} \to f^{(2)} .
\end{cases}
\tag{2.14}
$$

According to our convention, the field $C^{(2)}$ is a two-form $U(1)$ gauge field while $f^{(2)}$ is a

---

[4]For discrete 2-groups, see [6] for an approach in terms of topological defects.

[5]We emphasize that this gauging procedure renders the gauge field fully dynamical. This stands in contrast to gauging with flat connections, which is a topological manipulation that can be entirely encoded in changing the topological boundary conditions for the symmetry TFT. See [51,52] for some recent examples of the latter procedure.

two-form $\mathbb{R}$ gauge field, and both are dynamical. The resulting Symmetry TFT is:

$$S_5 = \frac{i}{2\pi} \int_{X^{(5)}} -b^{(3)} \wedge dA^{(1)} - f^{(2)} \wedge dC^{(2)} + \frac{k_{\mathsf{AC^2}}}{4\pi} A^{(1)} \wedge f^{(2)} \wedge f^{(2)}$$
$$+ \frac{k_{\mathsf{A^2C}}}{4\pi} f^{(2)} \wedge A^{(1)} \wedge dA^{(1)} + \frac{k_{\mathsf{A^3}}}{12\pi} A^{(1)} \wedge dA^{(1)} \wedge dA^{(1)} \,. \tag{2.15}$$

This action is invariant under the following gauge transformations:

$$f^{(2)} \to f^{(2)} + d\lambda_f^{(1)} \,, \quad A^{(1)} \to A^{(1)} + d\lambda_A^{(0)} \,,$$

$$b^{(3)} \to b^{(3)} + d\lambda_b^{(2)} + \frac{k_{\mathsf{AC^2}}}{2\pi} f^{(2)} \wedge \lambda_f^{(1)} + \frac{k_{\mathsf{AC^2}}}{4\pi} d\lambda_f^{(1)} \wedge \lambda_f^{(1)} + \frac{k_{\mathsf{A^2C}}}{4\pi} \lambda_f^{(1)} \wedge dA^{(1)} \,, \tag{2.16}$$

$$C^{(2)} \to C^{(2)} + d\lambda_C^{(1)} + \frac{k_{\mathsf{AC^2}}}{2\pi} \lambda_A^{(0)} (f^{(2)} + d\lambda_f^{(1)}) + \frac{k_{\mathsf{AC^2}}}{2\pi} \lambda_f^{(1)} \wedge A^{(1)} - \frac{k_{\mathsf{A^2C}}}{4\pi} d\lambda_A^{(0)} \wedge A^{(1)} \,.$$

The gauge transformation of $C^{(2)}$ is well-defined under composition of gauge transformations only if $k_{\mathsf{A^2C}} \in 2\mathbb{Z}$. Recalling that $k_{\mathsf{A^2C}}$ and $k_{\mathsf{AC^2}}$ must have the same parity, see Appendix A, the constraint on $k_{\mathsf{A^2C}}$ also implies that $k_{\mathsf{AC^2}} \in 2\mathbb{Z}$. Note also that the term proportional to $k_{\mathsf{AC^2}}$ does not spoil the quantization of $\int dC^{(2)}$ because it is real valued.

Let us now describe all the gauge invariant topological operators characterizing the symmetry TFT. For this, we consider the equations of motion, which can be written in the following way:

$$dA^{(1)} = 0 \,,$$
$$db^{(3)} - \frac{k_{\mathsf{AC^2}}}{4\pi} f^{(2)} \wedge f^{(2)} = 0 \,,$$
$$dC^{(2)} - \frac{k_{\mathsf{AC^2}}}{2\pi} f^{(2)} \wedge A^{(1)} = 0 \,, \tag{2.17}$$
$$df^{(2)} = 0 \,,$$

where some equations have been used to simplify the others. From the above equations,

we define the following four gauge-invariant topological operators:

$$W_n(\gamma^{(1)}) = \exp\left(in \int_{\gamma^{(1)}} A^{(1)}\right), \tag{2.18}$$

$$V_\alpha(\Pi^{(3)}, \Omega^{(4)}) = \exp\left(\frac{i\alpha}{2\pi} \int_{\Pi^{(3)}} b^{(3)} - \frac{i\alpha}{8\pi^2} \int_{\Omega^{(4)}} f^{(2)} \wedge \left(k_{\mathsf{AC^2}} f^{(2)} + k_{\mathsf{A^2C}} dA^{(1)}\right)\right), \tag{2.19}$$

$$T_m(\Sigma^{(2)}, \Omega^{(3)}) = \exp\left(im \int_{\Sigma^{(2)}} C^{(2)} - \frac{im}{4\pi} \int_{\Omega^{(3)}} A^{(1)} \wedge \left(2k_{\mathsf{AC^2}} f^{(2)} + k_{\mathsf{A^2C}} \, dA^{(1)}\right)\right), \tag{2.20}$$

$$U_\beta(\Sigma^{(2)}) = \exp\left(\frac{i\beta}{2\pi} \int_{\Sigma^{(2)}} f^{(2)}\right). \tag{2.21}$$

Following the terminology of [53], the operators $W_n$ and $U_\beta$ are genuine topological operators while $V_\alpha$ and $T_m$ are, in general, non-genuine. Let us briefly address why there are terms appearing in (2.19) and (2.20) that are trivialized by the equations of motion. When $k_{\mathsf{AC^2}} = 0$, the inclusion of these terms ensures that the topological operators remain gauge invariant off-shell, albeit at the cost of rendering them non-genuine. Note however that, when $k_{\mathsf{AC^2}} \neq 0$, the gauge invariance of (2.19) and (2.20) is only guaranteed on-shell. This implies that correlation functions with multiple operator insertions involving $V_\alpha$ and $T_m$ may require the presence of other non-gauge invariant operators to define a globally gauge invariant configuration, as we will see instantly.

The operators above have the following expected action on each other:

$$\left\langle W_n(\gamma^{(1)}) V_\alpha(\Pi^{(3)}, \Omega^{(4)}) \right\rangle = \exp\left(in\alpha \operatorname{Link}(\gamma^{(1)}, \Pi^{(3)})\right), \tag{2.22}$$

$$\left\langle T_m(\Sigma_1^{(2)}, \Omega^{(3)}) U_\beta(\Sigma_2^{(2)}) \right\rangle = \exp\left(im\beta \operatorname{Link}(\Sigma_1^{(2)}, \Sigma_2^{(2)})\right), \tag{2.23}$$

where $\operatorname{Link}(M^{(p)}, M^{(q)})$ denotes a 2-component link (defined in Appendix B) between the manifolds $M^{(p)}$ and $M^{(q)}$. When $M^{(p)}$ and $M^{(q)}$ are even-dimensional (as for $T$ and $U$), it is antisymmetric.

We also have non-standard relations when some operators cross each other. Due to the cubic interaction terms proportional to $k_{\mathsf{AC^2}}$ in (2.15), the operators $T_m$ and $V_\alpha$ satisfy:

$$\left\langle V_\alpha(\Pi_1^{(3)}, \Omega_1^{(4)}) T_m(\Sigma^{(2)}, \Omega_2^{(3)}) \right\rangle = \left\langle V_\alpha(\widetilde{\Pi}_1^{(3)}, \widetilde{\Omega}_1^{(4)}) T_m(\Sigma^{(2)}, \Omega_2^{(3)}) U_{m\alpha k_{\mathsf{AC^2}}}(\widetilde{\Omega}_1^{(4)} \cap \Omega_2^{(3)}) \right\rangle, \tag{2.24}$$

$$\left\langle T_n(\Sigma_1^{(2)}, \Omega_1^{(3)}) T_m(\Sigma_2^{(2)}, \Omega_2^{(3)}) \right\rangle = \left\langle T_n(\widetilde{\Sigma}_1^{(2)}, \widetilde{\Omega}_1^{(3)}) T_m(\Sigma_2^{(2)}, \Omega_2^{(3)}) W_{nmk_{\mathsf{AC^2}}}(\widetilde{\Omega}_1^{(3)} \cap \Omega_2^{(3)}) \right\rangle, \tag{2.25}$$

where on the left hand side we assumed that $\Omega_1 \cap \Omega_2 = 0$. On the right-hand side, we used a superscript $\sim$ to describe manifolds obtained from a smooth deformation of the

corresponding manifolds appearing on the left-hand side. In (2.24) and (2.25), the operators $U_{m\alpha k_{\mathsf{A}^2\mathsf{C}}}$ and $W_{nmk_{\mathsf{A}^2\mathsf{C}}}$ are typically defined on open manifolds terminating on $V_\alpha$ and $T_m$ and are therefore not gauge invariant. However, the presence of these open operators exactly cancels the non-trivial off-shell gauge transformations of (2.19) and (2.20). Therefore, the correlation functions are gauge-invariant despite involving manifestly non gauge-invariant operators.

Finally, cubic interaction terms proportional to $k_{\mathsf{A}^2\mathsf{C}}$ and $k_{\mathsf{A}^3}$ both give rise to non-trivial phases in correlation functions involving insertions of three operators. For instance we have

$$\left\langle V_{\alpha_1}(\Pi_1^{(3)})V_{\alpha_2}(\Pi_2^{(3)})T_m(\Sigma^{(2)})\right\rangle = \exp\left(i\widehat{k}_{\mathsf{A}^2\mathsf{C}}\alpha_1\alpha_2 m\,\mathrm{Link}(\Pi_1^{(3)},\Pi_2^{(3)},\Sigma^{(2)})_1\right)\,, \qquad (2.26)$$

where $\widehat{k}_{\mathsf{A}} \equiv \frac{1}{4\pi}k_{\mathsf{A}^2\mathsf{C}}$ and $\mathrm{Link}(\Pi_1^{(3)},\Pi_2^{(3)},\Sigma^{(2)})_1$ is the triple linking number of type 1.[6] (For the definition of generalized linking numbers, see appendix B.) By a similar argument, we also have that

$$\left\langle V_{\alpha_1}(\Pi_1^{(3)})V_{\alpha_2}(\Pi_2^{(3)})V_{\alpha_3}(\Pi_3^{(3)})\right\rangle = \exp\left(\frac{ik_{\mathsf{A}^3}}{4\pi^2}\alpha_1\alpha_2\alpha_3\,\mathrm{Link}(\Pi_1^{(3)},\Pi_2^{(3)},\Pi_3^{(3)})_2\right)\,, \qquad (2.28)$$

where $\Pi_1^{(3)},\Pi_2^{(3)},\Pi_3^{(3)}$ are contractible 3-manifolds and $\mathrm{Link}(\Pi_1^{(3)},\Pi_2^{(3)},\Pi_3^{(3)})_2$ is the triple linking number of type 2.

From (2.22) and (2.23) one gathers that $V_{2\pi}$ acts trivially on all $W_n$ operators, and similarly $U_{2\pi}$ acts trivially on $T_m$ operators. For $U_{2\pi}$, this trivial action extends to all operators, and hence it can be recognized as the identity operator, implying that $\beta$ is a $2\pi$-periodic parameter. For $V_{2\pi}$, there is a subtlety. While (2.24) is still compatible with it being the identity, the relations (2.26) and (2.28) imply that $V_{2\pi}$ can still implement a non-trivial action on $T_m$ or other $V_\alpha$ operators whenever $k_{\mathsf{A}^2\mathsf{C}} \neq 0$ and $k_{\mathsf{A}^3} \neq 0$. As a consequence, the parameter $\alpha$ of $V_\alpha$ is truly $2\pi$-periodic if and only if $k_{\mathsf{A}^2\mathsf{C}} = k_{\mathsf{A}^3} = 0$, otherwise the symmetry of the bulk theory generated by $V_\alpha$ is $\mathbb{R}$.

---

[6]More explicitly, the phase on the right-hand-side can be written as

$$\exp\left(im\alpha_1\alpha_2\widehat{k}_{\mathsf{A}^2\mathsf{C}}\int \delta^{(2)}(\Omega^{(3)})\wedge\left(\delta^{(1)}(\Omega_1^{(4)})\wedge d\delta^{(1)}(\Omega_2^{(4)}) + \delta^{(1)}(\Omega_2^{(4)})\wedge d\delta^{(1)}(\Omega_1^{(4)})\right)\right)\,. \qquad (2.27)$$

## 2.3. Boundary Conditions and Genuine Operators

Within the standard symmetry TFT framework, the collection of topological operators (2.18)-(2.21), along with appropriate boundary conditions on the dynamical 5d bulk fields, will generate both topological symmetry operators and non-topological charged operators in the boundary four-dimensional theory that enjoys the generalized Abelian global symmetry structure which we are describing. This section aims to outline this process. We will consistently examine the 5d Symmetry TFT (2.15) on a five-manifold $X^{(5)}$ featuring two distinct boundaries: one, denoted as $\mathcal{M}_P$, where the physical theory resides, and the other, denoted as $\mathcal{M}_T$, where we impose the topological boundary conditions for the bulk dynamical fields.

The set of operators that can terminate on $\mathcal{M}_T$ is determined by a choice of topological boundary conditions. Once the Symmetry TFT bulk is shrunk, a charged operator of the boundary physical theory on $\mathcal{M}_P$ is typically considered genuine if it is attached to a genuine bulk operator that can terminate on $\mathcal{M}_T$. We will actually slightly relax this condition, by allowing in the configuration above also for non-genuine bulk operators whose non-genuine part trivializes on $\mathcal{M}_T$.

To impose consistent boundary conditions on $\mathcal{M}_T$, two operators that can terminate topologically on $\mathcal{M}_T$ must act trivially on each other. Therefore, the allowed boundary conditions are constrained by the relations (2.22)-(2.28). To ensure that the symmetries of the physical theory are faithful, we must require that any operator acting trivially on all operators terminating on $\mathcal{M}_T$ must itself be able to terminate on $\mathcal{M}_T$. Consequently, the operators allowed to terminate on $\mathcal{M}_T$ must constitute a maximal set of mutually transparent operators: these operators form a Lagrangian algebra of the TQFT.

Let us now suppose that $M \in \mathbb{Z}$ is the minimal non-trivial charge at which $T_M$ can terminate on $\mathcal{M}_T$. From (2.23) it follows that $U_{2\pi/M}$ acts trivially on $T_M$, so $U_{2\pi/M}$ must also be trivialized on $\mathcal{M}_T$. Likewise, suppose that $N \in \mathbb{Z}$ is the minimal non-trivial charge at which $W_N$ can terminate on $\mathcal{M}_T$. Then, by (2.22), also $V_{2\pi/N}$ must be trivial on $\mathcal{M}_T$. Two operators terminating on $\mathcal{M}_T$ may generate new operators through (2.24) and (2.25). These new operators must be trivialized by the boundary conditions on $\mathcal{M}_T$ implying that $M$ and $N$ must satisfy the following constraint:

$$\frac{M^2}{N} k_{\mathsf{AC}^2} \in \mathbb{Z}\,. \tag{2.29}$$

Finally, to trivialize the triple linking relations (2.26) and (2.28) with insertions of $V_{2\pi/N}$

and $T_M$ operators, we require the additional constraints:

$$\frac{M}{N^2} k_{\mathsf{A}^2\mathsf{C}} \in 2\mathbb{Z}\,,$$
$$\frac{1}{N^3} k_{\mathsf{A}^3} \in \mathbb{Z}\,. \tag{2.30}$$

From a boundary four-dimensional theory perspective, taking $|N| \neq 1$ corresponds to gauging a $\mathbb{Z}_N^{(0)}$ subgroup of the $U(1)_\mathsf{A}^{(0)}$ symmetry. Likewise, taking $|M| \neq 1$ corresponds to gauging a $\mathbb{Z}_M^{(1)}$ subgroup of the magnetic $U(1)_\mathsf{C}^{(1)}$ symmetry. In what follows, we will solely consider the particular case where $N = M = 1$, i.e., we do not gauge any discrete subgroup of the $U(1)_\mathsf{A}^{(0)}$ and $U(1)_\mathsf{C}^{(1)}$ symmetries.

### 2.3.1. $U(1)_\mathsf{A}^{(0)} \times U(1)_\mathsf{C}^{(0)}$ without Mixed Anomalies

Let us first consider a four-dimensional theory with global symmetry $U(1)_\mathsf{A}^{(0)} \times U(1)_\mathsf{C}^{(0)}$ without mixed anomalies. In this case, gauging $U(1)_\mathsf{C}^{(0)}$ in four dimensions leads to a five-dimensional Symmetry TFT obtained from (2.15) by setting $k_{\mathsf{A}\mathsf{C}^2} = k_{\mathsf{A}^2\mathsf{C}} = 0$:

$$S_5 = \frac{i}{2\pi} \int -b^{(3)} \wedge dA^{(1)} - f^{(2)} \wedge dC^{(2)} + \frac{k_{\mathsf{A}^3}}{12\pi} A^{(1)} \wedge dA^{(1)} \wedge dA^{(1)}\,. \tag{2.31}$$

In this case, the bulk operators defined in (2.18)–(2.21) are all genuine operators. Boundary conditions trivializing the operators $W_1$ and $T_1$ when they lie on $\mathcal{M}_T$ must implement

$$\int_{\gamma^{(1)} \subset \mathcal{M}_T} A^{(1)} \in 2\pi\mathbb{Z}\,, \qquad \int_{\Sigma^{(2)} \subset \mathcal{M}_T} C^{(2)} \in 2\pi\mathbb{Z}\,. \tag{2.32}$$

One way to achieve this is by inserting the following boundary action on $\mathcal{M}_T$:

$$S_{\mathcal{M}_T} = \frac{i}{2\pi} \int_{\mathcal{M}_T} A^{(1)} \wedge d\widetilde{\Phi}^{(2)} + C^{(2)} \wedge dA_f^{(1)}\,, \tag{2.33}$$

where $\widetilde{\Phi}^{(2)}$ and $A_f^{(1)}$ are $U(1)$ gauge fields defined on $\mathcal{M}_T$. It is the sum over their fluxes that yields the quantization of the holonomies in (2.32). They can also be seen as edge modes for $b^{(3)}$ and $f^{(2)}$ respectively. Indeed, considering the terms proportional to $\delta A^{(1)}$ and $\delta C^{(2)}$ in the boundary variation of the total action:

$$\delta(S_5 + S_{\mathcal{M}_T})|_{\mathcal{M}_T} = 0\,, \tag{2.34}$$

we find the following boundary equations of motion:

$$b^{(3)}|_{\mathcal{M}_T} = d\widetilde{\Phi}^{(2)}, \quad f^{(2)}|_{\mathcal{M}_T} = dA_f^{(1)}. \tag{2.35}$$

Consistently with the Lagrangian algebra discussed above, these equations imply that precisely $V_{2\pi n}$ and $U_{2\pi m}$ are also trivialized when they lie on $\mathcal{M}_T$ for all $n, m \in \mathbb{Z}$.

Alternatively, the quantization conditions (2.32) can be implemented by introducing $U(1)$ edge modes for $A^{(1)}$ and $C^{(2)}$ as follows

$$A^{(1)}|_{\mathcal{M}_T} = d\Phi^{(0)}, \quad C^{(2)}|_{\mathcal{M}_T} = dA_C^{(1)}. \tag{2.36}$$

To make these edge modes appear explicitly in the boundary action, we can replace (2.33) with the following action:

$$S_{\mathcal{M}_T} = \frac{i}{2\pi} \int_{\mathcal{M}_T} b^{(3)} \wedge \left( d\Phi^{(0)} - A^{(1)} \right) - f^{(2)} \wedge \left( dA_C^{(1)} - C^{(2)} \right). \tag{2.37}$$

In this formulation, the sum over the fluxes of the edge modes implies that $V_{2\pi n}$ and $U_{2\pi m}$ are trivial on $\mathcal{M}_T$. Of course, we have also the option of choosing a different presentation of the boundary actions in each of the two sectors, which so far do not communicate.

Finally, the boundary constraints (2.35) and (2.36) are gauge invariant if we consider the following gauge transformations for the two sets of edge modes:

$$
\begin{aligned}
b^{(3)} &\to b^{(3)} + d\lambda_b^{(2)}, & f^{(2)} &\to f^{(2)} + d\lambda_f^{(1)}, \\
A^{(1)} &\to A^{(1)} + d\lambda_A^{(0)}, & C^{(2)} &\to C^{(2)} + d\lambda_C^{(1)}, \\
\widetilde{\Phi}^{(2)} &\to \widetilde{\Phi}^{(2)} + d\lambda_{\widetilde{\Phi}}^{(1)} + \lambda_b^{(2)}, & A_f^{(1)} &\to A_f^{(1)} + d\lambda_{A_f}^{(0)} + \lambda_f^{(1)}, \\
\Phi^{(0)} &\to \Phi^{(0)} + 2\pi n_\Phi + \lambda_A^{(0)}, & A_C^{(1)} &\to A_C^{(1)} + d\lambda_{A_C}^{(0)} + \lambda_C^{(1)}.
\end{aligned}
\tag{2.38}
$$

### 2.3.2. Non-Invertible $U(1)_\mathsf{A}^{(0)}$ Symmetry

In this subsection we will analyze a four dimensional theory in which the $U(1)_\mathsf{A}^{(0)} \times U(1)_\mathsf{C}^{(0)}$ global symmetry has a non-vanishing 't Hooft anomaly coefficient $k_{\mathsf{AC}^2} \neq 0$ while $k_{\mathsf{A}^2\mathsf{C}} = 0$. As discussed in section 2.1, gauging $U(1)_\mathsf{C}^{(0)}$ in such a theory will give rise to an ABJ anomaly for the $U(1)_\mathsf{A}^{(0)}$ symmetry. Although the Symmetry TFT in this case is a straightforward generalization of the theory presented in [19], here we provide additional technical details that were not included in the above reference but are crucial for the further generalizations discussed here.

Contrarily to the anomaly-free case, the bulk operators $T_m$ and $V_\alpha$ defined in (2.20) and (2.19) respectively are now non-genuine operators. We expect $T_m$ to terminate on genuine line operators charged under the $U(1)_C^{(1)}$ symmetry of the physical theory but this is only possible if $T_m$ is itself a genuine operator, since the non-genuine part in (2.20) cannot be trivialized by topological boundary conditions. Likewise, we want $V_\alpha$ to be identified to a genuine symmetry operators of the $U(1)_A^{(0)}$ symmetry. In both cases, these operators can be rendered genuine by stacking them with a suitable TQFT.

Let us first consider $T_m$ operators. A genuine version of these operators can be defined as follows:

$$T_m(\Sigma^{(2)}) = \exp\left(im \int_{\Sigma^{(2)}} C^{(2)} + \mathcal{T}_{2d}^{mk_{AC^2}}[A^{(1)}, f^{(2)}]\right), \qquad (2.39)$$

where $\mathcal{T}_{2d}$ is a 2d $U(1)/\mathbb{R}$ mixed BF theory coupled to the bulk fields $A^{(1)}$ and $f^{(2)}$:

$$\mathcal{T}_{2d}^p[A^{(1)}, f^{(2)}] = \frac{i}{2\pi} \int_{\Sigma^{(2)}} \phi^{(1)} \wedge d\Upsilon^{(0)} + p\Upsilon^{(0)} f^{(2)} + \phi^{(1)} \wedge A^{(1)}. \qquad (2.40)$$

Here, $\Upsilon^{(0)}$ and $\phi^{(1)}$ are respectively a dynamical $U(1)$ scalar and a one-form $\mathbb{R}$ gauge field.[7] The genuine operator $T_m$ is gauge invariant if, alongside the bulk transformations (2.16), we consider the following non-trivial gauge transformations for $\Upsilon^{(0)}$ and $\phi^{(1)}$:

$$\Upsilon^{(0)} \to \Upsilon^{(0)} + 2\pi n_\Upsilon - \lambda_A^{(0)}, \quad \phi^{(1)} \to \phi^{(1)} + d\lambda_\phi^{(0)} - mk_{AC^2}\lambda_f^{(1)}, \qquad (2.41)$$

where $n_\Upsilon \in \mathbb{Z}$. In particular, the action (2.40) is invariant under large gauge transformation of $\Upsilon^{(0)}$ because the bulk equations of motion imply $\int f^{(2)} \in 2\pi\mathbb{Z}$.

This 2d TQFT possesses non-genuine topological operators:

$$\exp\left(i\frac{\beta}{2\pi} \int_{\gamma^{(1)}} \phi^{(1)} + i\frac{\beta}{2\pi} mk_{AC^2} \int_{\Sigma_\gamma^{(2)}} f^{(2)}\right), \quad \exp\left(in(\Upsilon^{(0)}(\mathcal{P}_f) - \Upsilon^{(0)}(\mathcal{P}_i)) + in \int_{\mathcal{P}_i}^{\mathcal{P}_f} A^{(1)}\right), \qquad (2.42)$$

with $\partial\Sigma_\gamma^{(2)} = \gamma^{(1)}$. The non-genuine parts of these operators are open bulk $U_\beta$ and $W_n$ operators that typically extend outside of $\Sigma^{(2)}$, and that by this mechanism can end topologically on $T_m$ for any $\beta \in \mathbb{R}$ and $n \in \mathbb{Z}$, as required for consistency with (2.24) and (2.25).

Let us now consider the topological operator $V_\alpha(\Pi^{(3)}, \Omega^{(4)})$ from (2.19). Its non-genuine

---

[7]This choice of TQFT is not unique. We could for example insert the coefficient $p$ in the third term of (2.40) instead of the second one, as considered in [19]. However, the present choice of TQFT leads to a simple 't Hooft line in the physical theory.

part involves the term:

$$-\frac{i\alpha}{8\pi^2}k_{\mathsf{AC}^2}\int_{\Omega^{(4)}}f^{(2)}\wedge f^{(2)}\,. \tag{2.43}$$

Since $f^{(2)}$ satisfies the quantization condition $\int f^{(2)}\in 2\pi\mathbb{Z}$, when $\Omega^{(4)}$ is a closed spin 4-manifold, $V_\alpha(\Pi^{(3)},\Omega^{(4)})$ does not depend on $\Omega^{(4)}$ provided $\alpha k_{\mathsf{AC}^2}\in 2\pi\mathbb{Z}$. For these values of $\alpha$, $V_\alpha$ implements an invertible $\mathbb{Z}^{(0)}_{k_{\mathsf{AC}^2}}$ symmetry on the physical boundary. For all the other values of $\alpha$, the integral (2.43) can be replaced by a 3d TQFT supported on $\Pi^{(3)}=\partial\Omega^{(4)}$ whose action is [19]:

$$\mathcal{T}^{\alpha k_{\mathsf{AC}^2}}[f^{(2)}](\Pi^{(3)})=\frac{i}{2\pi}\int_{\Pi^{(3)}}-\frac{\alpha k_{\mathsf{AC}^2}}{4\pi}c^{(1)}\wedge dc^{(1)}+\Phi^{(1)}(dc^{(1)}+f^{(2)})\,, \tag{2.44}$$

where $c^{(1)}$ is a $\mathbb{R}$ gauge field while $\Phi^{(1)}$ is a $U(1)$ gauge field. Assigning to the latter the following gauge transformations:

$$c^{(1)}\to c^{(1)}+d\lambda_c^{(0)}-\lambda_f^{(1)}\,,\quad \Phi^{(1)}\to\Phi^{(1)}+d\lambda_\Phi^{(0)}-\frac{\alpha k_{\mathsf{AC}^2}}{2\pi}\lambda_f^{(1)}\,, \tag{2.45}$$

we can then define the following genuine, topological and gauge invariant operators:

$$V_\alpha(\Pi^{(3)})=\exp\left(\frac{i\alpha}{2\pi}\int_{\Pi^{(3)}}b^{(3)}+\mathcal{T}^{\alpha k_{\mathsf{AC}^2}}[f^{(2)}]\right)\,. \tag{2.46}$$

The above operators closely resemble to, and indeed allow us to recover, the non-invertible symmetry defects $\mathcal{D}_\alpha(\Pi^{(3)})$ of the four dimensional physical theory, as described in (2.10).

Using the definitions of genuine topological operators given in equations (2.39) and (2.46), we can analyze the mutual action of $T$ and $V$ and compare the results with the non-trivial relations (2.24)–(2.25). Note for instance that open $U_\beta$ operators can end on $V_\alpha$ defects by embodying the non-genuine part of a line operator for $\Phi^{(1)}$ in the 3d TQFT, see Appendix C for further details.

With the genuine bulk operators now properly defined, we can turn our attention to the topological boundary conditions. First of all, to account for the additional cubic interaction term in the bulk action, the boundary conditions discussed in Section 2.3.1 need to be modified. For this purpose we introduce the following boundary action:

$$S_{\mathcal{M}_T}=\frac{i}{2\pi}\int_{\mathcal{M}_T}b^{(3)}\wedge\left(d\Phi^{(0)}-A^{(1)}\right)+C^{(2)}\wedge dA_f^{(1)}-\frac{k_{\mathsf{AC}^2}}{4\pi}\Phi^{(0)}dA_f^{(1)}\wedge dA_f^{(1)}\,. \tag{2.47}$$

The boundary equations of motion, including the sum over fluxes of the edge modes, lead

to the following expressions for the bulk fields when restricted on the boundary $\mathcal{M}_T$:

$$A^{(1)}|_{\mathcal{M}_T} = d\Phi^{(0)}, \quad b^{(3)}|_{\mathcal{M}_T} = d\widetilde{\Phi}^{(2)} + \frac{k_{\mathsf{AC}^2}}{4\pi}A_f^{(1)} \wedge dA_f^{(1)},$$

$$C^{(2)}|_{\mathcal{M}_T} = dA_C^{(1)} + \frac{k_{\mathsf{AC}^2}}{2\pi}\Phi^{(0)} \wedge dA_f^{(2)}, \quad f^{(2)}|_{\mathcal{M}_T} = dA_f^{(1)}. \tag{2.48}$$

The edge modes must now also have $k_{\mathsf{AC}^2}$-dependent terms in their gauge transformations:

$$\Phi^{(0)} \to \Phi^{(0)} + 2\pi n_\Phi + \lambda_A^{(0)}, \quad \widetilde{\Phi}^{(2)} \to \widetilde{\Phi}^{(2)} + d\lambda_{\widetilde{\Phi}}^{(1)} + \lambda_b^{(2)} - \frac{k_{\mathsf{AC}^2}}{4\pi}\lambda_f^{(1)} \wedge A_f^{(1)},$$

$$A_C^{(1)} \to A_C^{(1)} + d\lambda_{A_C}^{(0)} + \lambda_C^{(1)} - \frac{k_{\mathsf{AC}^2}}{2\pi}\Phi^{(0)}\lambda_f^{(1)} - n_\Phi k_{\mathsf{AC}^2}A_f^{(1)}, \quad A_f^{(1)} \to A_f^{(1)} + d\lambda_{A_f}^{(0)} + \lambda_f^{(1)}. \tag{2.49}$$

With these gauge transformations, the transformation of the boundary action cancels the boundary terms arising from the transformation of the bulk action, taking into account the relations (2.48).

The boundary condition on $A^{(1)}$ allows $W_1 = \exp(i \int A^{(1)})$ to terminate topologically on $\mathcal{M}_T$. In order to see that also all $T_m$ can terminate topologically on $\mathcal{M}_T$, we have to consider not only the non-trivial boundary condition (2.48) on $C^{(2)}$, but also the fact that the 2d TQFT defined on $T_m(\Sigma^{(2)})$ has appropriate boundary conditions. Taking $\partial\Sigma^{(2)} \subset \mathcal{M}_T$, we select the boundary conditions that trivialize all the local operators of the TQFT $\mathcal{T}_{2d}$, namely

$$\int (\phi^{(1)} + mk_{\mathsf{AC}^2}A_f^{(1)}) \in 2\pi\mathbb{Z}, \quad \Upsilon^{(0)} + \Phi^{(0)} \in 2\pi\mathbb{Z}, \tag{2.50}$$

where we have used the edge modes on $\mathcal{M}_T$ to write fully gauge invariant combinations. The fact that all $\exp\left(in(\Upsilon^{(0)} + \Phi^{(0)})\right)$ operators are trivial on $\partial\Sigma^{(2)}$ is a required feature since we want the $T_m$ operators to define simple line operators of the physical theory after slab compactification. In particular, any local topological operator living on this line operator should be a multiple of the identity.

It is then possible to find an expression for the topological junction operator between $T_m(\Sigma^{(2)})$ and $\mathcal{M}_T$ imposing the boundary conditions (2.50) on the dynamical fields of the operator $T_m$:

$$\exp\left(im\int_{\partial\Sigma^{(2)}}A_C^{(1)} + \frac{i}{2\pi}\int_{\partial\Sigma^{(2)}}(\Upsilon^{(0)} + \Phi^{(0)})(mk_{\mathsf{AC}^2}A_f^{(1)} - d\Psi^{(0)})\right), \tag{2.51}$$

where $\Psi^{(0)}$ is an edge mode for $\phi^{(1)}$.

### 2.3.3. $U(1)_A^{(0)} \times U(1)_C^{(0)}$ with 2-Group Coefficient

Let us now instead focus on a four dimensional theory whose $U(1)_A^{(0)} \times U(1)_C^{(0)}$ global symmetries have a non-vanishing 't Hooft anomaly coefficient $k_{A^2C} \neq 0$ while $k_{AC^2} = 0$. As discussed in section 2.1, gauging $U(1)_C^{(0)}$ in such a theory will give rise to a 2-group global symmetry.

Consider first how to impose the topological boundary conditions in presence of a non-vanishing $k_{A^2C}$. We do so by introducing the following boundary action on $\mathcal{M}_T$:

$$S_{\mathcal{M}_T} = \frac{i}{2\pi} \int_{\mathcal{M}_T} b^{(3)} \wedge (d\Phi^{(0)} - A^{(1)}) + \left( C^{(2)} + \frac{k_{A^2C}}{4\pi} d\Phi^{(0)} \wedge A^{(1)} \right) \wedge dA_f^{(1)} \,, \qquad (2.52)$$

so that the bulk fields end up satisfying the following simple boundary constraints (we are using $k_{A^2C} \in 2\mathbb{Z}$):

$$\begin{aligned} A^{(1)}|_{\mathcal{M}_T} = d\Phi^{(0)} \,, \quad & b^{(3)}|_{\mathcal{M}_T} = d\widetilde{\Phi}^{(2)} \,, \\ C^{(2)}|_{\mathcal{M}_T} = dA_C^{(1)} \,, \quad & f^{(2)}|_{\mathcal{M}_T} = dA_f^{(1)} \,. \end{aligned} \qquad (2.53)$$

These boundary conditions are gauge invariant provided that we consider the associated gauge transformations for the edge modes:

$$\begin{aligned} \Phi^{(0)} \to \Phi^{(0)} + 2\pi n_\Phi + \lambda_A^{(0)} \,, \quad & \widetilde{\Phi}^{(2)} \to \widetilde{\Phi}^{(2)} + d\lambda_{\widetilde{\Phi}}^{(1)} + \lambda_b^{(2)} \,, \\ A_f^{(1)} \to A_f^{(1)} + d\lambda_{A_f}^{(0)} + \lambda_f^{(1)} \,, \quad & A_C^{(1)} \to A_C^{(1)} + d\lambda_{A_C}^{(0)} + \lambda_C^{(1)} - \frac{k_{A^2C}}{4\pi} \lambda_A^{(0)} d\Phi^{(0)} \,. \end{aligned} \qquad (2.54)$$

Turning to the definition of the extended operators of this theory, let us start with $V_\alpha$. We could define genuine bulk operators by stacking them with a TQFT as in the previous case. However, in the present physical theory, the $V_\alpha$ operators implement a $U(1)_A^{(0)}$ symmetry which is invertible, since $k_{AC^2} = 0$. Now, bulk $V_\alpha$ operators stacked with a TQFT are non-invertible, and they would fail to reproduce the desired invertible $U(1)_A^{(0)}$ operators of the physical theory even when brought to the boundary. We then propose not requiring $V_\alpha$ to be genuine throughout the entire bulk, but only on the topological boundary $\mathcal{M}_T$.

In fact, the boundary conditions (2.53) imply that, when the non-genuine part of $V_\alpha$ lies on $\mathcal{M}_T$, it becomes trivial:

$$\int_{\Omega^{(4)} \subset \mathcal{M}_T} f^{(2)} \wedge dA^{(1)} = 0 \,, \qquad (2.55)$$

for any (open or closed) $\Omega^{(4)} \subset \mathcal{M}_T$. Using the above condition, $\Omega^{(4)}$ appearing in the

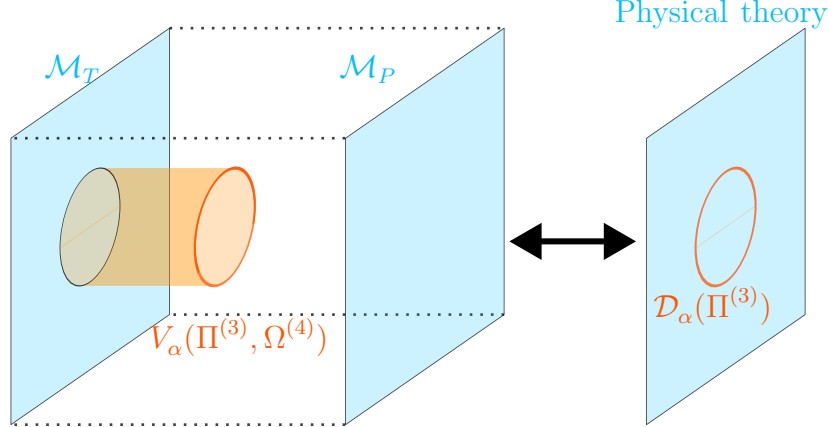

Figure 1: On the left: insertion of a $V_\alpha(\Pi^{(3)}, \Omega^{(4)})$ operator in the bulk. The 4-dimensional manifold $\Omega^{(4)}$ can be opened on $\mathcal{M}_T$. After a slab compactification, $V_\alpha(\Pi^{(3)}, \Omega^{(4)})$ becomes a genuine invertible $\mathcal{D}_\alpha(\Pi^{(3)})$ 0-form symmetry operator of the physical theory.

definition of $V_\alpha$ can be opened on $\mathcal{M}_T$ for any $\alpha \in \mathbb{R}$. In particular, if we choose $\partial\Omega^{(4)} = \Pi^{(3)}_{\mathcal{M}_T}$ with $\Pi^{(3)}_{\mathcal{M}_T} \subset \mathcal{M}_T$, we can consider

$$V_\alpha(\Pi^{(3)}_{\mathcal{M}_T}) = \exp\left(\frac{i\alpha}{2\pi} \int_{\Pi^{(3)}_{\mathcal{M}_T}} b^{(3)}\right), \tag{2.56}$$

which is a genuine topological operator. More generally, bulk $V_\alpha(\Pi^{(3)})$ operators can be attached to open cylinders of topology $\Pi^{(3)} \times I$, where $I$ denotes a closed interval, terminating on $\mathcal{M}_T$.[8] As discussed above, these operators are non-genuine in the bulk but become genuine at the topological boundary $\mathcal{M}_T$, see Fig. 1.

A very similar story applies to the operators $T_m$ defined in (2.20) which are non-genuine in the bulk. Their non-genuine part becomes trivial on the boundary:

$$\int_{\widetilde{\Omega}^{(3)} \subset \mathcal{M}_T} A^{(1)} \wedge dA^{(1)} = 0, \tag{2.57}$$

by virtue of the condition $A^{(1)} = d\Phi^{(0)}$ on $\mathcal{M}_T$. Furthermore, the junction operator between this non-genuine part and $\mathcal{M}_T$ is trivial. Being genuine, they can further open by attaching

---

[8]This is similar to the symmetry TFT of 3d Chern-Simons where bulk line operators are attached to cylinders ending on the boundary [54].

on a line operator of the edge mode $A_C^{(1)}$:

$$\exp\left(im\int_{\partial\Sigma^{(2)}} A_C^{(1)}\right).\tag{2.58}$$

Note that in principle, we could as well consider $T_m$ operators that are genuine also in the bulk, by replacing their non-genuine part by a $\mathcal{T}_{2d}^{\frac{m}{2}k_{\mathsf{A}^2\mathsf{C}}}[A^{(1)}, dA^{(1)}]$ 2d TQFT on $\Sigma^{(2)}$.[9] Provided we properly take care of the boundary conditions of the TQFT degrees of freedom, along the lines of the previous section, then these operators also give rise to simple 't Hooft lines on the boundary theory after slab compactification.

### 2.3.4. Most General $U(1)_{\mathsf{A}}^{(0)} \times U(1)_{\mathsf{C}}^{(0)}$ Symmetry Structure

We now finalize our analysis of the most general symmetry structures appearing in four dimensional QFTs with $U(1)_{\mathsf{A}}^{(0)} \times U(1)_{\mathsf{C}}^{(0)}$ global symmetries and consider the case where both 't Hooft anomaly coefficients $k_{\mathsf{A}^2\mathsf{C}}$ and $k_{\mathsf{A}\mathsf{C}^2}$ are not equal to zero.

To impose the topological boundary conditions on $\mathcal{M}_T$, we introduce a topological action including both cubic terms in (2.47) and (2.52):

$$S_{\mathcal{M}_T} = \frac{i}{2\pi}\int_{\mathcal{M}_T} b^{(3)}\wedge(d\Phi^{(0)}-A^{(1)})+\left(C^{(2)}+\frac{k_{\mathsf{A}^2\mathsf{C}}}{4\pi}d\Phi^{(0)}\wedge A^{(1)}\right)\wedge dA_f^{(1)}-\frac{k_{\mathsf{A}\mathsf{C}^2}}{4\pi}\Phi^{(0)}dA_f^{(1)}\wedge dA_f^{(1)}.\tag{2.59}$$

The boundary conditions imposed by this action are:

$$A^{(1)}|_{\mathcal{M}_T} = d\Phi^{(0)},\quad b^{(3)}|_{\mathcal{M}_T} = d\widetilde{\Phi}^{(2)}+\frac{k_{\mathsf{A}\mathsf{C}^2}}{4\pi}A_f^{(1)}\wedge dA_f^{(1)},$$

$$C^{(2)}|_{\mathcal{M}_T} = dA_C^{(1)}+\frac{k_{\mathsf{A}\mathsf{C}^2}}{2\pi}\Phi^{(0)}\wedge dA_f^{(2)},\quad f^{(2)}|_{\mathcal{M}_T} = dA_f^{(1)},\tag{2.60}$$

whose associated edge modes gauge transformation are given by

$$\Phi^{(0)}\to\Phi^{(0)}+2\pi n_\Phi+\lambda_A^{(0)},\quad A_f^{(1)}\to A_f^{(1)}+d\lambda_{A_f}^{(0)}+\lambda_f^{(1)},$$

$$\widetilde{\Phi}^{(2)}\to\widetilde{\Phi}^{(2)}+d\lambda_{\widetilde{\Phi}}^{(1)}+\lambda_b^{(2)}-\frac{k_{\mathsf{A}\mathsf{C}^2}}{4\pi}\lambda_f^{(1)}\wedge A_f^{(1)},\tag{2.61}$$

$$A_C^{(1)}\to A_C^{(1)}+d\lambda_{A_C}^{(0)}+\lambda_C^{(1)}-\frac{k_{\mathsf{A}^2\mathsf{C}}}{4\pi}\lambda_A^{(0)}d\Phi^{(0)}-n_\Phi k_{\mathsf{A}\mathsf{C}^2}A_f^{(1)}-\frac{k_{\mathsf{A}\mathsf{C}^2}}{2\pi}\Phi^{(0)}\lambda_f^{(1)}.$$

With these transformations, the boundary term arising from the variation of the bulk action

---

[9]For instance, we need to do this if we want to wrap them around a 2-cycle which is not a boundary. In this case, the operators of the 2d TQFT will play a crucial role to reproduce the relations (2.26).

is canceled by the transformation of the action (2.59).

Concerning the operators of the theory, we see that in this general case the non-genuine sector of the $V_\alpha$ operators (2.19) consists of two terms. The term proportional to $k_{\mathsf{AC}^2}$ can be removed by introducing the 3d TQFT $\mathcal{T}^\alpha[f^{(2)}]$ exactly as described in subsection 2.3.2, with the consequence of making them non-invertible. The remaining term, proportional to $k_{\mathsf{A}^2\mathsf{C}}$, can be opened on $\mathcal{M}_T$ without introducing another TQFT, as in subsection 2.3.3. The $V_\alpha$ operators are then both non-invertible and non-genuine, as they are attached to $\mathcal{M}_T$ through a cylinder:

$$V_\alpha(\Pi^{(3)}, \Omega^{(4)}) = \exp\left( \frac{i\alpha}{2\pi} \int_{\Pi^{(3)}} b^{(3)} + \mathcal{T}^{\alpha k_{\mathsf{AC}^2}}[f^{(2)}] - \frac{i\alpha}{8\pi^2} k_{\mathsf{A}^2\mathsf{C}} \int_{\Omega^{(4)}} f^{(2)} \wedge dA^{(1)} \right) , \qquad (2.62)$$

where $\partial\Omega^{(4)} = \Pi^{(3)} \cup \Pi^{(3)}_{\mathcal{M}_T}$, with $\Pi^{(3)}_{\mathcal{M}_T} \subset \mathcal{M}_T$.

Looking at (2.20), we see that also the non-genuine part of $T_m$ consists of two terms. One option is to do as for $V_\alpha$, and define operators which are both dressed by a TQFT and non-genuine. However in this case, it is perhaps simpler to define a completely genuine version of these operators, since both terms can be replaced by a single 2d TQFT. This theory can be expressed as follows,

$$\begin{aligned}
\mathcal{T}^{ml}_{2d}\left[ A^{(1)}, \frac{k_{\mathsf{A}^2\mathsf{C}}}{2l} dA^{(1)} + \frac{k_{\mathsf{AC}^2}}{l} f^{(2)} \right] &= \\
\frac{i}{2\pi} \int_{\Sigma^{(2)}} \phi^{(1)} \wedge d\Upsilon^{(0)} + m\Upsilon^{(0)} &\left( \frac{k_{\mathsf{A}^2\mathsf{C}}}{2} dA^{(1)} + k_{\mathsf{AC}^2} f^{(2)} \right) + \phi^{(1)} \wedge A^{(1)} ,
\end{aligned} \qquad (2.63)$$

where $l \equiv \gcd(\frac{1}{2} k_{\mathsf{A}^2\mathsf{C}}, k_{\mathsf{AC}^2})$ appears because

$$\int_{\Sigma^{(2)}} \left( \frac{k_{\mathsf{A}^2\mathsf{C}}}{2} dA^{(1)} + k_{\mathsf{AC}^2} f^{(2)} \right) \in 2\pi l \mathbb{Z} , \qquad (2.64)$$

for any closed 2-manifold $\Sigma^{(2)}$. Clearly, when $k_{\mathsf{AC}^2} = 0$ or $k_{\mathsf{A}^2\mathsf{C}} = 0$, we recover $\mathcal{T}^{\frac{m}{2} k_{\mathsf{A}^2\mathsf{C}}}_{2d}[A^{(1)}, dA^{(1)}]$ and $\mathcal{T}^{m k_{\mathsf{AC}^2}}_{2d}[A^{(1)}, f^{(2)}]$ respectively.

The various relations among these $V_\alpha$ and $T_m$ operators can be argued, as in the previous sections, to reproduce relations equivalent to (2.24)–(2.26). As for the boundary conditions on the dynamical fields of $\mathcal{T}^{ml}_{2d}$ on $\mathcal{M}_T$, they can be obtained with the same arguments as in Section 2.3.2. In particular, the line operator living at the junction between $T_m$ and $\mathcal{M}_T$ is still (2.51).

## 2.4. Scheme Dependence of the Anomaly Coefficients

A peculiar feature of four-dimensional theories with an Abelian 2-group global symmetry as in (2.5) is that their cubic 't Hooft anomaly coefficient $k_{A^3}$ is not completely scheme independent [4]. Thanks to the symmetry TFT introduced in (2.15), we can easily analyze how the anomaly coefficients can be further shifted in the presence of a non-trivial ABJ coefficient $k_{AC^2}$.

The basic idea is that, under a field redefinition:

$$f^{(2)} \to f^{(2)} + \ell dA^{(1)} , \tag{2.65}$$

with $\ell \in \mathbb{Z}$, the action (2.15) acquires the additional terms

$$\frac{i}{2\pi} \int_{X^{(5)}} -\ell dA^{(1)} \wedge dC^{(2)} + \frac{2\ell k_{AC^2}}{4\pi} f^{(2)} \wedge A^{(1)} \wedge dA^{(1)} + \frac{3\ell(\ell k_{AC^2} + k_{A^2C})}{12\pi} A^{(1)} \wedge dA^{(1)} \wedge dA^{(1)} , \tag{2.66}$$

The first term is a total derivative whose integral over closed manifolds is always in $2\pi i\mathbb{Z}$, it is therefore trivial in the bulk. The remaining terms induce the following shifts of the anomaly coefficients:

$$k_{A^2C} \to k_{A^2C} + 2\ell k_{AC^2} , \quad k_{A^3} \to k_{A^3} + 3\ell(\ell k_{AC^2} + k_{A^2C}) . \tag{2.67}$$

Since $k_{A^2C}$ and $k_{AC^2}$ have the same parity (see Appendix A), the term $3\ell(\ell k_{AC^2} + k_{A^2C})$ is always an integer multiple of 6. Moreover, $2\ell k_{AC^2}$ is always even, therefore the shift of $k_{A^2C}$ does not change its parity and it still satisfies $k_{A^2C} = k_{AC^2} \mod 2$.

Note that the shifts (2.67) highlight a new effect which was absent in theories with only 2-group global symmetry, namely that the presence of a non-invertible symmetry can induce a 2-group coefficient. From the physical theory perspective, this can be seen as redefining transformations of parameter $\alpha$ of $U(1)_A^{(0)}$ as a combination of a $U(1)_A^{(0)}$ transformation and a gauge transformation of the $U(1)_C^{(0)}$ gauge symmetry of parameter $l\alpha$. The charges $q_i^A$ becomes $q_i^A + lq_i^C$ and the anomaly coefficients computed with the new charges are

$$k'_{AC^2} = k_{AC^2} , \quad k'_{A^2C} = k_{A^2C} + 2\ell k_{AC^2} , \quad k'_{A^3} = k_{A^3} + 3\ell(\ell k_{AC^2} + k_{A^2C}) , \tag{2.68}$$

where we used $k_{C^3} = 0$.

# 3. Examples

In this section, we explore the generalized Abelian symmetry structure of the four-dimensional theories described in Section 2 by examining a few simple and explicit models. We will follow a strategy inspired by [4], namely we first consider a so called "parent" theory denoted by $T_1$ with $U(1)_A^{(0)} \times U(1)_C^{(0)}$ global symmetry and non-vanishing 't Hooft anomaly coefficients $k_{A^2C}$ and $k_{AC^2}$ to produce a "daughter" theory $T_2$ obtained from gauging the $U(1)_C^{(0)}$ symmetry. We will then discuss how to deform the theory $T_2$ while preserving the generalized symmetry structure, to understand how $T_2$ evolves under the RG flow. We first consider a gapless example corresponding to a vacuum where all the symmetry structure is spontaneously broken, and then a gapped example where the unbroken symmetry structure is encoded by a topologically ordered low-energy phase.

## 3.1. Gapless Example

It turns out that for both our examples, we can start with a parent theory $T_1$ consisting of four fermions and a scalar. The key point is to assign appropriate charges under $U(1)_A^{(0)} \times U(1)_C^{(0)}$ such that $k_{A^2C}$ and $k_{AC^2}$ are non-zero, while retaining $k_{C^3} = 0$. This is necessary to obtain the daughter theory $T_2$. The scalar field then has a charge under $U(1)_A^{(0)}$ in order to break the symmetry and gap the fermions through Yukawa couplings, after it condenses. As for its charge under $U(1)_C^{(0)}$, it has to vanish in our first example, since we want the $U(1)_C$ photon to remain massless after condensation of the scalar.

Therefore, we consider a parent theory $T_1$ consisting of $N_f = 4$ massless Weyl fermions $\psi_\alpha^i$ and a complex scalar field $\phi$ transforming under $U(1)_C^{(0)} \times U(1)_A^{(0)}$ with the following charges:

| **Field** | $\psi_\alpha^1$ | $\psi_\alpha^2$ | $\psi_\alpha^3$ | $\psi_\alpha^4$ | $\phi$ |
|---|---|---|---|---|---|
| $U(1)_A^{(0)}$ | $q_1^A = 3$ | $q_2^A = 4$ | $q_3^A = 5$ | $q_4^A = -6$ | $q_\phi^A = 1$ |
| $U(1)_C^{(0)}$ | $q_1^C = 1$ | $q_2^C = -1$ | $q_3^C = 1$ | $q_4^C = -1$ | $q_\phi^C = 0$ |

The relevant 't Hooft anomaly coefficients in $T_1$ (see appendix A for details) are:[10]

$$
\begin{aligned}
&k_{\mathsf{C}^3} = k_{\mathsf{A}^3} = 0 \ , \\
&k_{\mathsf{A}\mathsf{C}^2} = 3 + 4 + 5 - 6 = 6 \ , \\
&k_{\mathsf{A}^2\mathsf{C}} = 9 - 16 + 25 - 36 = -18 \ .
\end{aligned}
\tag{3.1}
$$

Since the $k_{\mathsf{C}^3}$ anomaly vanishes we are allowed to gauge $U(1)_{\mathsf{C}}^{(0)} \to U(1)_C^{(0)}$ to obtain the daughter theory $T_2$ which is a simple example of chiral Abelian gauge theory possessing the generalized symmetry structure that we introduced in section 2.

We are interested in analyzing the dynamics of $T_2$ by deforming the model with the scalar field $\phi$. As long as $\phi$ has a mass term such that it does not acquire a VEV, $T_2$ is IR free. Moreover, its magnetic 1-form symmetry current $j_{\mathsf{C}}^{(2)}$ defined in (2.6) decouples and the theory flows to a model in which the entire generalized Abelian symmetry structure is unbroken, though realized rather trivially. We will not be interested in this phase.

A more interesting scenario can be obtained by adding a Goldstone-like potential for $\phi$ together with Yukawa couplings between $\phi$ and the fermions. When $\phi$ acquires a vacuum expectation value $\langle \phi \rangle = v$, the Yukawa interactions can be used to give mass to all the fermions. The VEV also spontaneously breaks the $U(1)_{\mathsf{A}}^{(0)}$ symmetry in $T_1$, so that the phase of $\phi$ will become a Goldstone boson.

Given the above charges, a symmetry-preserving Yukawa interaction is for instance

$$
\mathscr{L}_{\mathrm{Y}} = \lambda_{12} \overline{\phi}^7 \psi^1 \psi^2 + \lambda_{34} \phi \, \psi^3 \psi^4 + (\mathrm{c.c.}) \, .
\tag{3.2}
$$

Other choices are possible but they will not affect our discussion. We should also not be concerned by the high dimension of one of the terms appearing above, as this toy model is not intended to be UV complete.

When the scalar field $\phi$ acquires a VEV, thus spontaneously breaking the $U(1)_{\mathsf{A}}^{(0)}$ symmetry in $T_1$, the terms in (3.2) give a mass to all the fermions. The $U(1)_{\mathsf{C}}^{(0)}$ symmetry on the other hand remains unbroken. The IR of $T_1$ is thus described by a Nambu-Goldstone boson for $U(1)_{\mathsf{A}}^{(0)}$ together with suitable terms to match both $k_{\mathsf{A}^2\mathsf{C}}$ and $k_{\mathsf{A}\mathsf{C}^2}$ anomalies. To obtain $T_2$ we now need to gauge $U(1)_{\mathsf{C}}^{(0)} \to U(1)_C^{(0)}$ and obtain a model, known as Goldstone-Maxwell, describing the spontaneously broken abelian symmetry structure that we identified in this work.[11]

---

[10]This theory also has a $k_{\mathsf{A}\mathscr{P}^2}$ mixed anomaly [4], we will not be concerned by it here.

[11]Similar theories, with $k_{\mathsf{A}\mathsf{C}^2} = 0$, have also been discussed in the context of axion physics [45–47].

The Goldstone-Maxwell model can be obtained directly from the Symmetry TFT by choosing appropriate boundary conditions, not only at the topological boundary, but also at the physical one. For this, we will explicitly write boundary actions on $\mathcal{M}_P$ and $\mathcal{M}_T$ imposing boundary conditions on the dynamical bulk fields. Performing a slab compactification leads to the physical theory corresponding to this symmetry TFT setup. The choice of boundary action on $\mathcal{M}_P$ is strongly informed by the discussions in [55], though with a slightly different perspective, that we will comment upon at the end of this section.

Let us recall the bulk action of the symmetry TFT which is given by (2.15) where we set $k_{\mathsf{A}^3} = 0$:

$$
\begin{aligned}
S_5 = \frac{i}{2\pi} \int_{X^{(5)}} &-b^{(3)} \wedge dA^{(1)} - f^{(2)} \wedge dC^{(2)} + \frac{k_{\mathsf{AC}^2}}{4\pi} A^{(1)} \wedge f^{(2)} \wedge f^{(2)} \\
&+ \frac{k_{\mathsf{A}^2\mathsf{C}}}{4\pi} f^{(2)} \wedge A^{(1)} \wedge dA^{(1)} \,.
\end{aligned}
\tag{3.3}
$$

We then introduce on $\mathcal{M}_T$ the topological boundary action (2.59) allowing the operators $W_1$, $T_1$, $V_{2\pi}$ and $U_{2\pi}$ to terminate topologically on this boundary.

On $\mathcal{M}_P$, we consider the following boundary action:

$$
\begin{aligned}
S_{\mathcal{M}_P} = \frac{R^2}{4\pi} \int_{\mathcal{M}_P} &\left( A^{(1)} - \mathsf{A}^{(1)} \right) \wedge * \left( A^{(1)} - \mathsf{A}^{(1)} \right) \\
&+ \int_{\mathcal{M}_P} \frac{i}{2\pi} \left[ C^{(2)} - \mathsf{C}^{(2)} + \frac{k_{\mathsf{A}^2\mathsf{C}}}{4\pi} \mathsf{A}^{(1)} \wedge \mathsf{A}^{(1)} \right] \wedge dY^{(1)} + \frac{1}{2e^2} dY^{(1)} \wedge *dY^{(1)} \,,
\end{aligned}
\tag{3.4}
$$

where $Y^{(1)}$ is an edge mode on $\mathcal{M}_P$, while $\mathsf{A}^{(1)}$ and $\mathsf{C}^{(2)}$ are background gauge fields also defined there. To obtain a gauge invariant boundary, we need to fix the gauge of $b^{(3)}$ and $f^{(2)}$ on $\mathcal{M}_P$ and consider only gauge transformations where $\lambda_b^{(2)}$ and $\lambda_f^{(1)}$ vanish on $\mathcal{M}_P$. On the other hand, the background fields $\mathsf{A}^{(1)}$ and $\mathsf{C}^{(2)}$ allow us to consider non-trivial gauge transformations of $A^{(1)}$ and $C^{(2)}$ on $\mathcal{M}_P$. We consider the following gauge transformations for the boundary fields:

$$
Y^{(1)} \to Y^{(1)} + d\lambda_Y^{(0)}, \quad \mathsf{A}^{(1)} \to \mathsf{A}^{(1)} + d\lambda_A^{(0)}, \quad \mathsf{C}^{(2)} \to \mathsf{C}^{(2)} + d\lambda_C^{(1)} - \frac{k_{\mathsf{A}^2\mathsf{C}}}{4\pi} d\lambda_A^{(0)} \wedge \mathsf{A}^{(1)} \,.
\tag{3.5}
$$

However, when $k_{\mathsf{AC}^2} \neq 0$ introducing background fields is not enough to obtain gauge invariance on $\mathcal{M}_P$. Indeed one gets

$$
\delta(S_{\mathcal{M}_P} + S_5)|_{\mathcal{M}_P} = \frac{-ik_{\mathsf{AC}^2}}{8\pi^2} \int_{\mathcal{M}_P} \lambda_A^{(0)} f^{(2)} \wedge (f^{(2)} - 2dY^{(1)}) \,.
\tag{3.6}
$$

The boundary $\mathcal{M}_P$ is gauge invariant only if this additional term is in $2\pi\mathbb{Z}$. This implies that we need to restrict the gauge transformations to parameters satisfying $k_{\mathsf{AC}^2}\lambda_A^{(0)} \in 2\pi\mathbb{Z}$, meaning we can only consider gauge transformations for a $\mathbb{Z}_{k_{\mathsf{AC}^2}}^{(0)}$ subgroup of $U(1)_\mathsf{A}^{(0)}$. This is not a surprise, since we only expect a well-defined coupling to a background gauge field for invertible symmetry transformations, which is precisely the case for $\mathbb{Z}_{k_{\mathsf{AC}^2}}^{(0)} \subset U(1)_\mathsf{A}^{(0)}$. We will nevertheless continue to use general $\lambda_A^{(0)}$ in the following, as they help unveil also the non-invertible part of the symmetry structure.

The boundary action (3.4) imposes the following conformal boundary conditions:

$$b^{(3)} - \frac{k_{\mathsf{A}^2\mathsf{C}}}{4\pi}f^{(2)} \wedge (A^{(1)} - \mathsf{A}^{(1)}) = -iR^2 * (A^{(1)} - \mathsf{A}^{(1)}),$$

$$d\left[C^{(2)} - \mathsf{C}^{(2)} + \frac{k_{\mathsf{A}^2\mathsf{C}}}{4\pi}\mathsf{A}^{(1)} \wedge A^{(1)} - i\frac{2\pi}{e^2} * dY^{(1)}\right] = 0, \quad f^{(2)} = dY^{(1)}. \tag{3.7}$$

These boundary conditions are gauge invariant, provided the following subtlety. In the expression between brackets the following term does not cancel:

$$\delta C^{(2)} \supset \frac{k_{\mathsf{AC}^2}}{2\pi}\lambda_A^{(0)}f^{(2)} = nf^{(2)}, \tag{3.8}$$

where we used $k_{\mathsf{AC}^2}\lambda_A^{(0)} = n \in \mathbb{Z}$. Since the boundary condition imposes $df^{(2)} = 0$, this extra term is eventually annihilated by the exterior derivative.

Now, the constraint on $C^{(2)}$ can be solved as follows:

$$C^{(2)} - \mathsf{C}^{(2)} + \frac{k_{\mathsf{A}^2\mathsf{C}}}{4\pi}\mathsf{A}^{(1)} \wedge A^{(1)} - \frac{i2\pi}{e^2} * dY^{(1)} = d\widetilde{Y}^{(1)}. \tag{3.9}$$

The term $\frac{k_{\mathsf{AC}^2}}{2\pi}\lambda_A^{(0)}f^{(2)}$ arising from the gauge transformation of $C^{(2)}$ is not annihilated by an exterior derivative anymore, so that gauge invariance of (3.9) requires the following gauge transformation for $\widetilde{Y}^{(1)}$:

$$\widetilde{Y}^{(1)} \to \widetilde{Y}^{(1)} + d\lambda_{\widetilde{Y}}^{(0)} + \frac{k_{\mathsf{AC}^2}}{2\pi}\lambda_A^{(0)}Y^{(1)}. \tag{3.10}$$

Since both $Y^{(1)}$ and $\widetilde{Y}^{(1)}$ are $U(1)$ fields, this gauge transformation is only well defined if $k_{\mathsf{AC}^2}\lambda_A^{(0)} \in 2\pi\mathbb{Z}$, i.e., if we only consider transformations belonging to the $\mathbb{Z}_{k_{\mathsf{AC}^2}}^{(0)}$ (invertible) subgroup of $U(1)_\mathsf{A}^{(0)}$.

Let us now shrink the bulk to get an on-shell action which is given by $S_{\mathrm{GM}} = S_{\mathcal{M}_P} -$

$S_{\mathcal{M}_T}$.[12] Using the expressions (2.59) and (3.4), and substituting all the bulk fields with edge modes, we get

$$
\begin{aligned}
S_{\text{GM}} =& \frac{R^2}{4\pi} \int \left( d\Phi^{(0)} - \mathsf{A}^{(1)} \right) \wedge * \left( d\Phi^{(0)} - \mathsf{A}^{(1)} \right) + \frac{1}{2e^2} \int dA_f^{(1)} \wedge *dA_f^{(1)} \\
&+ \frac{ik_{\mathsf{AC}^2}}{8\pi^2} \int \Phi^{(0)} \wedge dA_f^{(1)} \wedge dA_f^{(1)} + \frac{ik_{\mathsf{A}^2\mathsf{C}}}{8\pi^2} \int \mathsf{A}^{(1)} \wedge d\Phi^{(0)} \wedge dA_f^{(1)} - \frac{i}{2\pi} \int \mathsf{C}^{(2)} \wedge dA_f^{(1)} ,
\end{aligned}
$$
(3.11)

where we have further identified $dA_f^{(1)} = dY^{(1)}$ since they are both equated to $f^{(2)}$.

The first three terms in $S_{\text{GM}}$ correspond to the action of the Goldstone-Maxwell model and the last two are couplings to background fields. Note that the Goldstone-Maxwell theory possesses more symmetries than the ones described by the symmetry TFT.[13] Indeed the theory (3.11) has an emergent $U(1)_w^{(2)}$ winding symmetry whose current is $*\frac{1}{2\pi}d\Phi^{(0)}$ and an emergent electric $U(1)_e^{(1)}$ symmetry due to the dynamical $U(1)$ gauge field $A_f^{(1)}$. When $k_{\mathsf{AC}^2} \neq 0$, the latter symmetry is non-invertible.

Let us conclude this section with some comments on the differences and similarities between [55] and our approach. First, in order to describe a holographic duality between a bulk and a boundary theory, the work [55] introduces only a single boundary component and performs the gauging of a Lagrangian algebra inside the bulk. This procedure is equivalent to the slab compactification described in our work. Indeed, instead of gauging the Lagrangian algebra in the full bulk, we can first consider (half-)gauging in a subregion of the bulk. Gauging a Lagrangian algebra in a TQFT trivializes the TQFT, therefore, the half-gauging procedure creates a topological boundary between the bulk TQFT and a trivial theory. Such boundary corresponds to the topological boundary $\mathcal{M}_T$ in the symmetry TFT setup. The slab compactification corresponds to superimposing such topological boundary with the physical one, which can be interpreted as gauging the Lagrangian algebra of the full bulk, as described in [55].

We also note that, to study the 4d $U(1)_{\mathsf{A}}^{(0)}$ chiral anomaly, the authors of [55] freeze the gauge transformation of this symmetry by setting $\lambda_A^{(0)} = 0$ at the boundary. However, when $k_{\mathsf{AC}^2} \neq 1$, we expect the physical theory to exhibit an invertible $\mathbb{Z}_{k_{\mathsf{AC}^2}}^{(0)}$ symmetry with well-defined gauge transformations. Introducing the edge modes $Y^{(1)}$ and $\widetilde{Y}^{(1)}$, as we did in (3.4) and (3.9), allows us to preserve the $\mathbb{Z}_{k_{\mathsf{AC}^2}}^{(0)}$ gauge transformations. The necessity of

---

[12]Here we are taking into account the opposite orientation of the two boundaries of the slab.

[13]The symmetry TFT of Goldstone-Maxwell can be obtained from (2.15) by replacing the $U(1)$ fields $A^{(1)}$ and $C^{(2)}$ by $\mathbb{R}$ fields $a^{(1)}$ and $c^{(2)}$.

introducing these edge modes can be related to the Witten effect, as explained in the next subsection. Note that when $k_{\mathsf{A}\mathsf{C}^2} = 0$, the edge modes are not required and can consistently be integrated out. In this case, we recover the action used in [55] to analyze a 4d theory with higher-group symmetry.

### 3.1.1. Comments on 't Hooft Line Operators

In this subsection, we point out a subtlety concerning the 't Hooft lines in the Goldstone-Maxwell theory (3.11), and how our symmetry TFT set up provides the simplest fix.

Let us consider the insertion of a 't Hooft line $H_m(\gamma^{(1)})$. By definition, a 't Hooft line of charge $m$ is an operator imposing $\int_{\Sigma^{(2)}} dA_f^{(1)} = 2\pi m$ when integrated on a 2-cycle with linking number one with $\gamma^{(1)}$. Such operator can be defined as the boundary of the following operator:

$$H_m(\gamma^{(1)}) = \exp\left( 2\pi m \int_{\Sigma_\gamma^{(2)}} *j_e^{(2)} \right) , \tag{3.12}$$

where $\partial\Sigma_\gamma^{(2)} = \gamma^{(1)}$ and $*j_e^{(2)}$ is the current of the $U(1)_e^{(1)}$ symmetry shifting $A_f^{(1)}$ by a closed 1-form.[14] Extracting $j_e^{(2)}$ from (3.11), we therefore have

$$H_m(\gamma^{(1)}) = \exp\left( -\frac{2\pi m}{e^2} \int_{\Sigma_\gamma^{(2)}} *dA_f^{(1)} + im \int_{\Sigma_\gamma^{(2)}} \left( \mathsf{C}^{(2)} + \frac{k_{\mathsf{A}^2\mathsf{C}}}{4\pi} d\Phi^{(0)} \wedge \mathsf{A}^{(1)} - \frac{k_{\mathsf{A}\mathsf{C}^2}}{2\pi} \Phi^{(0)} \wedge dA_f^{(1)} \right) \right) . \tag{3.13}$$

From the above expression, it is easy to check that 't Hooft lines are not invariant under the gauge transformation $\Phi^{(0)} \to \Phi^{(0)} + 2\pi n_\Phi$, rather they transform as

$$H_m(\gamma^{(1)}) \to H_m(\gamma^{(1)}) \exp\left( -imn_\Phi k_{\mathsf{A}\mathsf{C}^2} \int_{\gamma^{(1)}} A_f^{(1)} \right) . \tag{3.14}$$

This fact was observed in [29] (see also [56]), where, in order to get a gauge invariant operator, it was suggested to insert a quantum mechanics (QM) on the t'Hooft line to compensate the phase.

From our symmetry TFT perspective, this QM arises from the stacking of $T_m$ with the two-dimensional TQFT $\mathcal{T}_{2d}$, as in (2.39) and (2.40). Let us reconsider the slab compactification for $\mathcal{T}_{2d}$: the topological boundary conditions on $\mathcal{M}_T$ are imposed by the boundary action appearing in (2.51).[15] On $\mathcal{M}_P$, as for the whole theory, we can choose to impose

---

[14]Despite involving $\Sigma_\gamma^{(2)}$ in its definition, $H_m(\gamma)$ is a genuine operator. Indeed, if $\Sigma^{(2)}$ is closed, the operator (3.12) implements a $U(1)_e^{(1)}$ transformation of parameter $2\pi m$. It is therefore trivial.

[15]To cancel the gauge transformation of $H_m$, this action must be inserted with the appropriate orientation.

the following conformal boundary condition:

$$*\Upsilon^{(0)} = iL\phi^{(1)}, \tag{3.16}$$

with $L$ a parameter with the dimension of length. This boundary condition can be imposed by inserting the following action at the junction between $T_m$ and $\mathcal{M}_P$

$$\frac{1}{4\pi L} \int_{\gamma^{(1)}} \Upsilon^{(0)} \wedge *\Upsilon^{(0)}. \tag{3.17}$$

After superimposing the two boundaries, the 't Hooft line is stacked with the following theory:

$$\frac{1}{4\pi L} \int_{\gamma^{(1)}} \Upsilon^{(0)} \wedge *\Upsilon^{(0)} + \frac{i}{2\pi} \int_{\gamma^{(1)}} \left(\Upsilon^{(0)} + \Phi^{(0)}\right) \left(mk_{\mathsf{AC}^2} A_f - d\Psi^{(0)}\right). \tag{3.18}$$

Integrating out $\Upsilon^{(0)}$ gives

$$\int_{\gamma^{(1)}} \frac{L}{4\pi} \left(d\Psi^{(0)} - mk_{\mathsf{AC}^2} A_f^{(1)}\right) \wedge * \left(d\Psi^{(0)} - mk_{\mathsf{AC}^2} A_f^{(1)}\right) - \frac{i}{2\pi} \Phi^{(0)} \left(d\Psi^{(0)} - mk_{\mathsf{AC}^2} A_f\right), \tag{3.19}$$

reproducing precisely the QM dressing proposed in [29], that leads to a simple 't Hooft line. The shift $\Phi^{(0)} \to \Phi^{(0)} + 2\pi n_\Phi$ inserts a Wilson line of charge $mn_\Phi k_{\mathsf{AC}^2}$ on the 't Hooft line canceling the one obtained in (3.14). This implies that the 't Hooft line coupled with the QM is invariant under the shift $\Phi^{(0)} \to \Phi^{(0)} + 2\pi n_\Phi$ and is therefore gauge invariant. Note finally that both the expression (3.13) and the boundary term (3.17) are not invariant under gauge transformations of parameter $\lambda_A^{(0)}$. However, in the physical theory, this is a transformation of a global symmetry, not a gauge transformation, so that it needs not be cancelled.

## 3.2. Gapped Example

We now move to study a different example where the scalar Higgs field $\phi$ is charged under the $U(1)_C$ gauge symmetry of $T_2$. As we will see, this drastically changes the IR

---

Using the boundary conditions on $\mathcal{M}_T$ and $\mathcal{M}_P$, the expression (3.13) can be written as

$$H_m(\gamma^{(1)}) = \exp\left(im \int_{\gamma^{(1)}} A_C^{(1)} - \widetilde{Y}^{(1)}\right). \tag{3.15}$$

Comparing the $A_C^{(1)}$ dependent part with the junction operator (2.51), this allows us to select the correct sign for the boundary action. Note that this expression for the 't Hooft line requires (3.10) to have the correct global symmetry transformation.

dynamics and leads to a gapped phase described by a topological quantum field theory.

Let us consider the following family of theories:

| **Field** | $\psi_\alpha^1$ | $\psi_\alpha^2$ | $\psi_\alpha^3$ | $\psi_\alpha^4$ | $\phi$ |
|---|---|---|---|---|---|
| $U(1)_{\mathsf{A}}^{(0)}$ | $q_1^{\mathsf{A}} = a$ | $q_2^{\mathsf{A}} = b$ | $q_3^{\mathsf{A}} = -a$ | $q_4^{\mathsf{A}} = -b$ | $q_\phi^{\mathsf{A}} = b - a$ |
| $U(1)_{\mathsf{C}}^{(0)}$ | $q_1^{\mathsf{C}} = x$ | $q_2^{\mathsf{C}} = -x$ | $q_3^{\mathsf{C}} = y$ | $q_4^{\mathsf{C}} = -y$ | $q_\phi^{\mathsf{C}} = y - x$ |

where $a, b, x, y$ are all integer numbers. The 't Hooft anomaly coefficients following from this charge assignment are:

$$
\begin{aligned}
k_{\mathsf{C}^3} &= k_{\mathsf{A}^3} = 0 \ , \\
k_{\mathsf{A}\mathsf{C}^2} &= -(b+a)(y+x)(y-x) \ , \\
k_{\mathsf{A}^2\mathsf{C}} &= -(b+a)(b-a)(y+x) \ .
\end{aligned}
\tag{3.20}
$$

Note that they satisfy the relation

$$
k_{\mathsf{A}^2\mathsf{C}} = \frac{q_\phi^{\mathsf{A}}}{q_\phi^{\mathsf{C}}} k_{\mathsf{A}\mathsf{C}^2} \ ,
\tag{3.21}
$$

which we will use below.

These theories allow the following symmetry-preserving Yukawa interactions:

$$
\mathscr{L}_{\mathrm{Y}} = \lambda_{14} \phi \psi^1 \psi^4 + \lambda_{23} \overline{\phi} \psi^2 \psi^3 + (\text{c.c.}) \ .
\tag{3.22}
$$

As in the previous section, when $\phi$ acquires a VEV $\langle\phi\rangle = v$, a real massless boson remains in the ungauged theory $T_1$, while the Yukawa interactions give a mass to all fermions in the theory. After going to $T_2$ by gauging, since $q_\phi^{\mathsf{C}} \neq 0$, the Goldstone boson is not invariant under transformations of the $U(1)_C$ gauge symmetry. The $U(1)_C$ gauge symmetry is therefore Higgsed to a $\mathbb{Z}_{q_\phi^{\mathsf{C}}}$ gauge symmetry. All dynamical fields in the theory are massive, so that we conclude that the $T_2$ theory is gapped and the non-trivial anomaly matching in the IR must be realized by a $\mathbb{Z}_{q_\phi^{\mathsf{C}}}$ TQFT with unbroken Abelian symmetry structure.[16]

---

[16]Note that the VEV of $\phi$ does not break the global symmetry arising from a transformations under $U(1)_{\mathsf{A}}^{(0)}$ that is compensated by a gauge transformation in $U(1)_C^{(0)}$. Another comment is that if $q_\phi^{\mathsf{C}} = 1$, then the IR is trivially gapped.

To obtain a TQFT as a physical theory upon shrinking the bulk of the Symmetry TFT, we need to consider topological boundary conditions on both boundaries. For simplicity, we restrict our study to the case where $q_\phi^{\mathsf{A}} = 1$.

On $\mathcal{M}_T$, we already discussed topological boundary conditions around equation (2.60). For later convenience, we will not implement them using the boundary action (2.59) but with an equivalent action (based on (2.33)) where the edge mode $\widetilde{\Phi}^{(2)}$ appears explicitly:

$$S_{\mathcal{M}_T} = \frac{i}{2\pi} \int_{\mathcal{M}_T} A^{(1)} \wedge \left( d\widetilde{\Phi}^{(2)} + \frac{k_{\mathsf{AC}^2}}{4\pi} A_f^{(1)} \wedge dA_f^{(1)} \right) + C^{(2)} \wedge dA_f^{(1)} \,. \tag{3.23}$$

The boundary conditions imposed by this action are:[17]

$$b^{(3)} = d\widetilde{\Phi}^{(2)} + \frac{k_{\mathsf{A}^2\mathsf{C}}}{4\pi} A^{(1)} \wedge dA_f^{(1)} + \frac{k_{\mathsf{AC}^2}}{4\pi} A_f^{(1)} \wedge dA_f^{(1)} \,, \quad f^{(2)} = dA_f^{(1)} \,. \tag{3.24}$$

These boundary conditions are gauge invariant if we consider:

$$\widetilde{\Phi}^{(2)} \to \widetilde{\Phi}^{(2)} + d\lambda_{\widetilde{\Phi}}^{(1)} + \lambda_b^{(2)} - \frac{k_{\mathsf{AC}^2}}{4\pi} \lambda_f^{(1)} \wedge A_f^{(1)} - \frac{k_{\mathsf{A}^2\mathsf{C}}}{4\pi} \left( \lambda_f^{(1)} \wedge A^{(1)} + \lambda_A^{(0)} d(A_f^{(1)} + \lambda_f^{(1)}) \right) \,,$$
$$A_f^{(1)} \to A_f^{(1)} + d\lambda_{A_f}^{(0)} + \lambda_f^{(1)} \,.$$
$$\tag{3.25}$$

Moreover, the transformation of the action $S_{\mathcal{M}_T}$ exactly cancels the boundary terms arising from the gauge transformation of the bulk action.

On the physical boundary $\mathcal{M}_P$, we should introduce an action resembling (3.4) but which is nevertheless topological. The action (3.4) can be equivalently written as

$$S_{\mathcal{M}_P} = \int_{\mathcal{M}_P} \frac{i}{2\pi} \left( A^{(1)} - \mathsf{A}^{(1)} \right) \wedge dX^{(2)} + \frac{1}{4\pi R^2} dX^{(2)} \wedge *dX^{(2)}$$
$$+ \int_{\mathcal{M}_P} \frac{i}{2\pi} \left[ C^{(2)} - \mathsf{C}^{(2)} + \frac{k_{\mathsf{A}^2\mathsf{C}}}{4\pi} \mathsf{A}^{(1)} \wedge A^{(1)} \right] \wedge dY^{(1)} + \frac{1}{2e^2} dY^{(1)} \wedge *dY^{(1)} \,.$$
$$\tag{3.26}$$

We can then replace the two non-topological kinetic terms for the edge modes $X^{(2)}$ and $Y^{(1)}$ with a topological BF term implementing the fact that we expect a $\mathbb{Z}_{q_\phi^{\mathsf{C}}}$ gauge theory

---

[17]Note that these boundary conditions are simply related to the one discussed in equation (2.60) through a field redefinition of $\widetilde{\Phi}^{(2)}$.

in this gapped vacuum:

$$S_{\mathcal{M}_P} = \frac{i}{2\pi} \int_{\mathcal{M}_P} (A^{(1)} - \mathsf{A}^{(1)}) \wedge dX^{(2)} - q_\phi^{\mathsf{C}} X^{(2)} \wedge dY^{(1)} + \left( C^{(2)} - \mathsf{C}^{(2)} + \frac{k_{\mathsf{A}^2\mathsf{C}}}{4\pi} \mathsf{A}^{(1)} \wedge A^{(1)} \right) \wedge dY^{(1)} .$$

(3.27)

This action imposes the following boundary conditions:

$$f^{(2)} = dY^{(1)} , \quad b^{(3)} = dX^{(2)} + \frac{k_{\mathsf{A}^2\mathsf{C}}}{4\pi} (A^{(1)} - \mathsf{A}^{(1)}) \wedge dY^{(1)} .$$

(3.28)

The boundary condition on $C^{(2)}$ can be written as follow:

$$C^{(2)} - \mathsf{C}^{(2)} + \frac{k_{\mathsf{A}^2\mathsf{C}}}{4\pi} \mathsf{A}^{(1)} \wedge A^{(1)} = d\widetilde{Y}^{(1)} + q_\phi^{\mathsf{C}} X^{(2)}$$

(3.29)

As in the gapless case, this constraint requires a non-trivial gauge transformation for $\widetilde{Y}^{(1)}$ when $k_{\mathsf{AC}^2} \neq 0$:

$$\widetilde{Y}^{(1)} \to \widetilde{Y}^{(1)} + d\lambda_{\widetilde{Y}}^{(0)} - q_\phi^{\mathsf{C}} \lambda_X^{(1)} + \frac{k_{\mathsf{AC}^2}}{2\pi} \lambda_A^{(0)} Y^{(1)} ,$$

(3.30)

since both fields are $U(1)$, this shift requires $k_{\mathsf{AC}^2} \lambda_A^{(0)} \in 2\pi\mathbb{Z}$, as in the gapless case.

As in the previous section, we fix the gauge of $b^{(3)}$ and $f^{(2)}$ on $\mathcal{M}_P$. This action and the boundary conditions are therefore gauge invariant if we consider the following gauge transformations for $X^{(2)}$ and $Y^{(1)}$:

$$X^{(2)} \to X^{(2)} + d\lambda_X^{(1)} , \quad Y^{(1)} \to Y^{(1)} + d\lambda_Y^{(0)} .$$

(3.31)

Upon shrinking the five dimensional bulk, we get the following action:

$$S_{\mathcal{M}_P} - S_{\mathcal{M}_T} = \frac{i}{2\pi} \int (A^{(1)} - \mathsf{A}^{(1)}) \wedge dX^{(2)} - q_\phi^{\mathsf{C}} X^{(2)} \wedge dY^{(1)} + (C^{(2)} - \mathsf{C}^{(2)} + \frac{k_{\mathsf{A}^2\mathsf{C}}}{4\pi} \mathsf{A}^{(1)} \wedge A^{(1)}) \wedge dY^{(1)}$$
$$- \frac{i}{2\pi} \int A^{(1)} \wedge \left( d\widetilde{\Phi}^{(2)} + \frac{k_{\mathsf{AC}^2}}{4\pi} A_f^{(1)} \wedge dA_f^{(1)} \right) + C^{(2)} \wedge dA_f^{(1)} .$$

(3.32)

After integrating out $A^{(1)}$ and $C^{(2)}$, we get immediately

$$S_{\mathrm{BF}} = -\frac{i}{2\pi} \int \mathsf{A}^{(1)} \wedge dX^{(2)} + q_\phi^{\mathsf{C}} X^{(2)} \wedge dY^{(1)} + \mathsf{C}^{(2)} \wedge dY^{(1)} .$$

(3.33)

Using the equations of motion, and the relation $k_{\mathsf{AC}^2} = q_\phi^{\mathsf{C}} k_{\mathsf{A}^2\mathsf{C}}$ that descends from (3.21),

we can equivalently write the action in terms of the other edge modes $\widetilde{\Phi}^{(2)}$ and $A_f^{(1)}$:

$$S_{\mathrm{BF}} = -\frac{i}{2\pi} \int \mathsf{A}^{(1)} \wedge d\widetilde{\Phi}^{(2)} + q_\phi^{\mathsf{C}} \widetilde{\Phi}^{(2)} \wedge dA_f^{(1)} + \mathsf{C}^{(2)} \wedge dA_f^{(1)} \,. \tag{3.34}$$

In both its variants, this is the characteristic four-dimensional BF action of a $\mathbb{Z}_{q_\phi^{\mathsf{C}}}^{(0)}$ gauge theory coupled to the background fields $\mathsf{A}^{(1)}$ and $\mathsf{C}^{(2)}$. The above action obtained from the symmetry TFT coincide with the one obtained in [4] when $k_{\mathsf{AC}^2} = 0$.

The anomaly and generalized Abelian symmetry structure of (3.33) and (3.34) are hidden inside the non-trivial gauge transformations of their background and dynamical fields. Indeed, let us first consider the following gauge transformation:

$$\mathsf{C}^{(2)} \to \mathsf{C}^{(2)} + d\lambda_C^{(1)} - \frac{k_{\mathsf{A}^2\mathsf{C}}}{4\pi} d\lambda_A^{(0)} \wedge \mathsf{A}^{(1)} \,. \tag{3.35}$$

Using the above transformation in (3.33), together with the standard ones for the other fields, we get

$$S_{\mathrm{BF}} \to S_{\mathrm{BF}} - \frac{ik_{\mathsf{A}^2\mathsf{C}}}{8\pi^2} \int \lambda_A^{(0)} \wedge d\mathsf{A}^{(1)} \wedge dY^{(1)} \,. \tag{3.36}$$

This is exactly the anomaly that leads to a 2-group, related to the conservation equation (2.4). Taking now the expression (3.34) for the same theory, and recalling that also $\widetilde{\Phi}^{(2)}$ still has a non-trivial transformation after setting $\lambda_f^{(1)} = 0 = \lambda_b^{(2)}$:

$$\widetilde{\Phi}^{(2)} \to \widetilde{\Phi}^{(2)} + d\lambda_{\widetilde{\Phi}}^{(1)} - \frac{k_{\mathsf{A}^2\mathsf{C}}}{4\pi} \lambda_A^{(0)} \wedge dA_f^{(1)} \,, \tag{3.37}$$

we see that the action transforms as

$$S_{\mathrm{BF}} \to S_{\mathrm{BF}} + \frac{ik_{\mathsf{AC}^2}}{8\pi^2} \int \lambda_A^{(0)} \wedge dA_f^{(1)} \wedge dA_f^{(1)} \,, \tag{3.38}$$

where we have used again the relation $k_{\mathsf{AC}^2} = q_\phi^{\mathsf{C}} k_{\mathsf{A}^2\mathsf{C}}$. This is now exactly what one expects in a theory with an ABJ anomaly, as in (2.8).

At first, it would seem that this theory has a different anomaly according to the variables in which one decides to write it. However as we already stressed, these two variants are equivalent. In fact, in this very simple topological theory, the two types of anomalies are two ways to see the same effect. Indeed, the equations of motion of $X^{(2)}$, or of $\widetilde{\Phi}^{(2)}$, imply that

$$d\mathsf{A}^{(1)} = -q_\phi^{\mathsf{C}} dY^{(1)} = -q_\phi^{\mathsf{C}} dA_f^{(1)} \,. \tag{3.39}$$

It is then straightforward to see that the two anomalies are equivalent on-shell.

Finally, we would like to stress that (as in the gapless example) the non-invariance of the theory under a gauge transformation with parameter $\lambda_A^{(0)}$ implies that $\mathsf{A}^{(1)}$ cannot be interpreted as a conventional background gauge field for $U(1)_\mathsf{A}^{(0)}$. It can however be rightfully considered as a background $\mathbb{Z}_{k_{\mathsf{AC}^2}}^{(0)}$ gauge field.

## 4. A Toy Model in Five Dimensions

In this section, we examine another example of a theory that exhibits both non-invertible and higher-group symmetries. This theory is remarkably simple, defined in five dimensions, with a single (vector) gauge field and one non-trivial topological interaction. We will observe that it reproduces many of the key features we encountered earlier in our four-dimensional discussion. One advantage of this example is that its symmetry structure exists inherently, without the need for any gauging. As in the four-dimensional case, we will frame much of our analysis in terms of a six-dimensional Symmetry TFT and its associated boundary conditions.

### 4.1. Maxwell-Chern–Simons Theory

Let us consider the following 5-dimensional Maxwell theory in Euclidean signature, to which we add a cubic Chern-Simons-like interaction term:

$$S[A^{(1)}] = \frac{1}{2} \int dA^{(1)} \wedge *dA^{(1)} - \frac{ik}{24\pi^2} \int A^{(1)} \wedge dA^{(1)} \wedge dA^{(1)} \ . \tag{4.1}$$

Note that the $U(1)$ Chern–Simons term can be written as

$$k\mathrm{CS}_5[A^{(1)}] = -2\pi \cdot \frac{k}{6(2\pi)^3} \int_{Y^{(6)}} F^{(2)} \wedge F^{(2)} \wedge F^{(2)} \,, \tag{4.2}$$

where $F^{(2)} = dA^{(1)}$ and $\partial Y^{(6)} = M^{(5)}$. On generic $Y^{(6)}$, $k$ must be an integer multiple of 6. If $Y^{(6)}$ is spin, the quantization condition can be relaxed and $k \in \mathbb{Z}$. Unless otherwise stated we will always assume that $Y^{(6)}$ is a spin manifold.

When $k = 0$, the theory is invariant under a $U(1)_\mathrm{e}^{(1)}$ electric global symmetry whose transformations act on $A^{(1)}$ as

$$A^{(1)} \to A^{(1)} + \lambda_\mathrm{e}^{(1)} \,, \tag{4.3}$$

with $\lambda_e^{(1)}$ a closed 1-form. The conserved current associated with this symmetry is

$$J_e^{(2)} = -dA^{(1)} \ . \tag{4.4}$$

When $k \neq 0$, the conservation law of this current becomes

$$d * J_e^{(2)} = -\frac{ik}{8\pi^2} dA^{(1)} \wedge dA^{(1)} \ . \tag{4.5}$$

The presence of the cubic interaction term breaks the electric 1-form symmetry $U(1)_e^{(1)}$ to $\mathbb{Z}_{k,e}^{(1)}$ whose associated symmetry defect can be defined as

$$D_\alpha(\Pi^{(3)}) = \exp\left( -\alpha \int_{\Pi^{(3)}} * J_e^{(2)} - \frac{i\alpha k}{8\pi^2} \int_{\Pi^{(3)}} A^{(1)} \wedge dA^{(1)} \right) , \tag{4.6}$$

where $\Pi^{(3)}$ is a closed three-manifold and the range of the angle $\alpha$ is restricted to $\alpha k \in 2\pi\mathbb{Z}$.

The electric 1-form symmetry has a 't Hooft anomaly obstructing its gauging whose inflow action is given by [57]:

$$S_6[\mathsf{B}_e^{(2)}] = -\frac{ik}{24\pi^2} \int_{Y^{(6)}} \mathsf{B}_e^{(2)} \wedge \mathsf{B}_e^{(2)} \wedge \mathsf{B}_e^{(2)} , \tag{4.7}$$

where $\mathsf{B}_e^{(2)}$ is a background field for $\mathbb{Z}_{k,e}^{(1)}$ whose background gauge transformation is $\mathsf{B}_e^{(2)} \to \mathsf{B}_e^{(2)} + d\lambda_e^{(1)}$.

The theory also possesses a $U(1)_m^{(2)}$ topological symmetry whose conserved current is

$$J_m^{(3)} = \frac{i}{2\pi} * dA^{(1)} , \tag{4.8}$$

and whose associated symmetry defect is

$$U_\alpha(\Sigma^{(2)}) = \exp\left( \frac{i\alpha}{2\pi} \int_{\Sigma^{(2)}} dA^{(1)} \right) , \tag{4.9}$$

with $\alpha \in [0, 2\pi)$. Note that the conservation law of magnetic 3-form current is not affected by the Chern–Simons interaction. The right-hand side of the anomalous conservation law (4.5) only involves the current of the $U(1)_m^{(2)}$ symmetry.

This situation is similar to the case $k_{\mathsf{AC}^2} \neq 0$ considered in Section 2 and the $U(1)_e^{(1)}$ symmetry can be fully restored. This is achieved by topologically higher-gauging the magnetic 2-form symmetry and interpreting it as a fully-fledged $U(1)_e^{(1)}$ non-invertible 1-form

global symmetry, in a simple adaptation of the results of [19]. The non-invertible $U(1)_{\mathrm{e}}^{(1)}$ topological operators are thus defined by

$$\mathcal{D}_\alpha(\Pi^{(3)}) = \exp\left(-\alpha \int_{\Pi^{(3)}} *J_e + \mathcal{T}^{k\alpha}[dA^{(1)}]\right), \tag{4.10}$$

where $\mathcal{T}^\alpha$ is the TQFT introduced in Section 2.1. The parameter $\alpha$ now takes values in $[0, 2\pi)$.[18]

One can further notice that five-dimensional Maxwell-Chern–Simons theory exhibits a higher-group global symmetry structure [28]. In what follows we will couple the magnetic symmetry to a corresponding 3-form background gauge field denoted by $\mathsf{B}_{\mathrm{m}}^{(3)}$. The most general higher-group invariant action coupled to all background gauge fields is given by:

$$S_6[A^{(1)}, \mathsf{B}_{\mathrm{e}}^{(2)}, \mathsf{B}_{\mathrm{m}}^{(3)}] \supset -\frac{ik}{24\pi^2} \int_{Y^{(6)}} (dA^{(1)} - \mathsf{B}_{\mathrm{e}}^{(2)}) \wedge (dA^{(1)} - \mathsf{B}_{\mathrm{e}}^{(2)}) \wedge (dA^{(1)} - \mathsf{B}_{\mathrm{e}}^{(2)})$$

$$-\frac{ik}{24\pi^2} \int_{Y^{(6)}} \mathsf{B}_{\mathrm{e}}^{(2)} \wedge \mathsf{B}_{\mathrm{e}}^{(2)} \wedge \mathsf{B}_{\mathrm{e}}^{(2)} + \frac{i}{2\pi} \int_{Y^{(6)}} \mathsf{H}^{(4)} \wedge (dA^{(1)} - \mathsf{B}_{\mathrm{e}}^{(2)}) .$$

The above action is independent of the choice of $Y^{(6)}$ if we postulate the following expression for the field strength $\mathsf{H}^{(4)}$:

$$\mathsf{H}^{(4)} = d\mathsf{B}_{\mathrm{m}}^{(3)} + \frac{k}{4\pi} \mathsf{B}_{\mathrm{e}}^{(2)} \wedge \mathsf{B}_{\mathrm{e}}^{(2)} . \tag{4.11}$$

For this quantity to be gauge invariant, we must impose the following gauge transformations:

$$\begin{cases} \mathsf{B}_{\mathrm{e}}^{(2)} \to \mathsf{B}_{\mathrm{e}}^{(2)} + d\lambda_{\mathrm{e}}^{(1)}, \\ \mathsf{B}_{\mathrm{m}}^{(3)} \to \mathsf{B}_{\mathrm{m}}^{(3)} + d\lambda_{\mathrm{m}}^{(2)} - \frac{k}{2\pi}\lambda_{\mathrm{e}}^{(1)} \wedge \mathsf{B}_{\mathrm{e}}^{(2)} - \frac{k}{4\pi}\lambda_{\mathrm{e}}^{(1)} \wedge d\lambda_{\mathrm{e}}^{(1)}, \end{cases} \tag{4.12}$$

where we have used the fact that $d\mathsf{B}_{\mathrm{e}}^{(2)} = 0$, since $\mathsf{B}_{\mathrm{e}}^{(2)}$ is a flat background. The symmetries, $\mathbb{Z}_{k,\mathrm{e}}^{(1)}$ and $U(1)_{\mathrm{m}}^{(2)}$ therefore constitute a non-trivial higher-group.

Finally, the theory admits a codimension-1 symmetry defect associated with the conserved current

$$J_I^{(1)} = *\frac{i}{8\pi^2} F^{(2)} \wedge F^{(2)}, \tag{4.13}$$

for an ordinary 0-form global symmetry $U(1)_I^{(0)}$ whose charged objects are $5d$ *instanton operators*. Furthermore we can always activate a background field $\mathsf{B}_I^{(1)}$ for $U(1)_I^{(0)}$ subject to the gauge transformation $\mathsf{B}_I^{(1)} \to \mathsf{B}_I^{(1)} + d\lambda_I^{(0)}$. However, the $U(1)_I^{(0)}$ symmetry does not

---

[18]A precursor of this analysis already appeared in [28]. Indeed, when $\alpha = 2\pi\frac{M}{N} \in 2\pi\mathbb{Q}$ with $\gcd(kM, N) = 1$, one can alternatively use the $\mathcal{A}^{N,kM}[dA^{(1)}]$ theory [58] instead of $\mathcal{T}^{\alpha k}[dA^{(1)}]$.

act faithfully on the pure Maxwell-CS theory we are considering since the $5d$ CS term implies that a (probe) instanton operator is not gauge invariant. For this reason, we will only have limited interest on the role of $U(1)_I^{(0)}$.[19]

## 4.2. Symmetry TFT Analysis

In this section we introduce a symmetry TFT to reproduce the generalized Abelian symmetry structure appearing in five-dimensional Maxwell-Chern–Simons (4.1). Let us start by writing down a six-dimensional action of the form:

$$S_6 = \frac{i}{2\pi} \int_{X^{(6)}} -b^{(3)} \wedge da^{(2)} + \frac{k}{12\pi} a^{(2)} \wedge a^{(2)} \wedge a^{(2)} \,, \tag{4.14}$$

where *both* $a^{(2)}$ and $b^{(3)}$ are $\mathbb{R}$ gauge fields and $k \in \mathbb{R}$. When $k = 0$, this theory corresponds to the symmetry TFT for a $U(1)^{(1)} \times U(1)^{(2)}$ symmetry with mixed anomaly [41], i.e., it describes Maxwell theory in five dimensions.

The action (4.14) is invariant under the following gauge transformations:

$$
\begin{aligned}
a^{(2)} &\to a^{(2)} + d\lambda_a^{(1)} \,, \\
b^{(3)} &\to b^{(3)} + d\lambda_b^{(2)} + \frac{k}{2\pi} \lambda_a^{(1)} \wedge a^{(2)} + \frac{k}{4\pi} \lambda_a^{(1)} \wedge d\lambda_a^{(1)} \,.
\end{aligned}
\tag{4.15}
$$

Since both $a^{(2)}$ and $b^{(3)}$ are $\mathbb{R}$ gauge fields, they can be freely rescaled. In particular, when $k \neq 0$, we can always choose a rescaling such that $k$ is set to 1. Therefore, we conclude that its value in the bulk is unphysical and in particular, not quantized. Nevertheless, we will keep a non-trivial $k$ because it will map to the Chern-Simons level appearing at the physical boundary, whose quantization will be fixed by the boundary conditions.

Let us now define the topological operators of the theory by analyzing first the bulk equations of motion:

$$da^{(2)} = 0 \,, \quad db^{(3)} - \frac{k}{4\pi} a^{(2)} \wedge a^{(2)} = 0 \,. \tag{4.16}$$

---

[19]Instanton operators and the $U(1)_I^{(0)}$ symmetry play a crucial role in the study of global symmetry enhancements in $5d$ supersymmetric gauge theories, see [59,60] and references therein for further details.

From each of these equations, we can define a topological gauge-invariant operator:

$$V_\alpha(\Pi^{(3)}, \Omega^{(4)}) = \exp\left(\frac{i\alpha}{2\pi}\left(\int_\Pi b^{(3)} - \frac{k}{4\pi}\int_\Omega a^{(2)} \wedge a^{(2)}\right)\right),\qquad (4.17)$$

$$U_\beta(\Sigma^{(2)}) = \exp\left(\frac{i\beta}{2\pi}\int_\Sigma a^{(2)}\right).\qquad (4.18)$$

These operators act on each other through linking:

$$\left\langle V_\alpha(\Pi^{(3)}, \Omega^{(4)}) U_\beta(\Sigma^{(2)})\right\rangle = \exp\left(\frac{i\alpha\beta}{2\pi}\mathrm{Link}(\Pi^{(3)}, \Sigma^{(2)})\right).\qquad (4.19)$$

From this relation, we see that $V_\alpha$ and $U_\beta$ are non-trivial topological operators when $\alpha, \beta \neq 0$, therefore $\alpha$ and $\beta$ take value in $\mathbb{R}$ and are not periodic parameters. Due to the cubic interaction term, we also have the relation

$$\left\langle V_\alpha(\Pi_1^{(3)}, \Omega_1^{(4)}) V_\beta(\Pi_2^{(3)}, \Omega_2^{(4)})\right\rangle = \left\langle V_\alpha(\widetilde{\Pi}_1^{(3)}, \widetilde{\Omega}_1^{(4)}) V_\beta(\Pi_2^{(3)}, \Omega_2^{(4)}) U_{\frac{k\alpha\beta}{2\pi}}(\widetilde{\Omega}_1^{(4)} \cap \Omega_2^{(4)})\right\rangle,\qquad (4.20)$$

where we assumed that $\Omega_1^{(4)} \cap \Omega_2^{(4)} = 0$ and we used $\widetilde{\Pi}_1^{(1)}$ and $\widetilde{\Omega}_1^{(4)}$ to denote manifolds obtained by continuously deforming $\Pi^{(1)}$ and $\Omega^{(4)}$.[20]

Note that the operators $V_\alpha$ defined above are non-genuine. As we did in the four-dimensional example, we might want to define their genuine counterpart. However, in the present example, the role played by bulk operators in the boundary theory will strongly depend on the boundary conditions. We will then assess the possibility of making the $V_\alpha$ operators genuine after we have discussed the boundary conditions.

### 4.3. Boundary Conditions and Slab Compactification

To define first topological boundary conditions on $\mathcal{M}_T$, we must select a maximal set of mutually transparent operators. The operators in this set can then terminate topologically at the boundary, meaning that they are trivialized by the boundary conditions.

Let us first quickly discuss what happens for $k = 0$. Looking at (4.19), we see that $V_\alpha$ and $U_\beta$ act trivially on each other when $\alpha r^{-1} \in 2\pi\mathbb{Z}$ and $\beta r \in 2\pi\mathbb{Z}$, for any $r \in \mathbb{R}$. Therefore, choosing a value of $r$ is equivalent to choosing a boundary condition. Rescaling the real bulk fields also changes the value of this parameter. In what follows, we will assume that $r = 1$ to avoid clutter. As we discussed below (4.15), note that this choice of

---

[20]Note that $\widetilde{\Omega}_1^{(4)} \cap \Omega_2^{(4)}$ can be non-trivial even if $\widetilde{\Pi}_1^{(3)} = \Pi_1^{(3)}$. In general, $\widetilde{\Omega}_1^{(4)} \cap \Omega_2^{(4)}$ is an open manifold terminating on $\widetilde{\Pi}_1^{(3)}$ and $\Pi_2^{(3)}$.

normalization fixes the value $k$ of the cubic interaction coupling in the bulk. We will now see that such $k$ is quantized and corresponds to the Chern-Simons level of the physical boundary theory.

When $k \neq 0$, due to (4.20), bulk $V_\alpha$ operators may not be transparent to themselves. More specifically, $V_{2\pi}$ is transparent to itself if and only if $k \in \mathbb{Z}$:

$$\left\langle V_{2\pi}(\Pi_1^{(3)}, \Omega_1^{(4)}) V_{2\pi}(\Pi_2^{(3)}, \Omega_2^{(4)}) \right\rangle = \left\langle V_{2\pi}(\widetilde{\Pi}_1^{(3)}, \widetilde{\Omega}_1^{(4)}) V_{2\pi}(\Pi_2^{(3)}, \Omega_2^{(4)}) U_{2\pi k}(\Omega_2^{(4)} \cap \widetilde{\Omega}_1^{(4)}) \right\rangle . \quad (4.21)$$

The operator generated by two $V_{2\pi}$ operators is $U_{2\pi k}$, which is trivialized on $\mathcal{M}_T$ by the boundary conditions if $k \in \mathbb{Z}$. Therefore, the quantization condition on the boundary Chern-Simons level is enforced by the consistency of the symmetry TFT boundary conditions.

Let us now consider the following boundary conditions for $a^{(2)}$ and $b^{(2)}$:

$$a^{(2)}|_{\mathcal{M}_T} = dA^{(1)} , \quad b^{(3)}|_{\mathcal{M}_T} = d\widetilde{A}^{(2)} + \frac{k}{4\pi} A^{(1)} \wedge dA^{(1)} , \quad (4.22)$$

where $A^{(1)}$ and $\widetilde{A}^{(2)}$ are $U(1)$ gauge fields. Taking inspiration from (2.47), these boundary conditions can be imposed by adding the following boundary action on $\mathcal{M}_T$:

$$S_{\mathcal{M}_T} = \frac{i}{2\pi} \int b^{(3)} \wedge (dA^{(1)} - a^{(2)}) - \frac{k}{12\pi} A^{(1)} \wedge dA^{(1)} \wedge dA^{(1)} . \quad (4.23)$$

The boundary conditions are gauge invariant if we consider

$$A^{(1)} \to A^{(1)} + d\lambda_A^{(0)} + \lambda_a^{(1)} , \quad \widetilde{A}^{(2)} \to \widetilde{A}^{(2)} + d\lambda_{\widetilde{A}}^{(1)} - \frac{k}{4\pi}(\lambda_A^{(0)} dA^{(1)} + \lambda_a^{(1)} \wedge A^{(1)} + \lambda_A^{(0)} d\lambda_a^{(1)}) , \quad (4.24)$$

and the gauge transformation of (4.23) cancels the boundary term arising from the bulk action.

Let us now consider the conditions at the physical boundary $\mathcal{M}_P$. The action of the physical theory can be obtained by considering the following boundary action on $\mathcal{M}_P$:

$$S_{\mathcal{M}_P} = -\frac{1}{2} \int a^{(2)} \wedge *a^{(2)} . \quad (4.25)$$

This choice imposes conformal boundary conditions:

$$b^{(3)} = -2\pi i * a^{(2)} . \quad (4.26)$$

When the two boundaries $\mathcal{M}_P$ and $\mathcal{M}_T$ are superimposed, we get a 5-dimensional theory whose action is given by the difference of the boundary actions (4.23) and (4.25). Expressing the bulk fields in terms of the edge modes, we then get

$$S = \frac{1}{2} \int dA^{(1)} \wedge *dA^{(1)} - \frac{ik}{24\pi^2} \int A^{(1)} \wedge dA^{(1)} \wedge dA^{(1)} \,, \tag{4.27}$$

which reproduces exactly the action of the theory we started with in (4.1).

### 4.4. Operators from the Symmetry TFT to the Physical Theory

We now discuss how the bulk operators $V_\alpha$ (4.17) and $U_\beta$ (4.18) reproduce both the symmetry defects and the charged operators of the boundary theory, given the topological boundary conditions we have imposed. Notably, we must address the fact that $V_\alpha$ is non-genuine in the bulk, while we expect both the symmetry defects and the charged operators to which it reduces on the boundary to be genuine.

Before addressing this issue, let us consider the simpler case of $U_\beta$. The boundary condition on $a^{(2)}$ from (4.22) implies that

$$\oint_{\Sigma^{(2)}} a^{(2)} \in 2\pi\mathbb{Z} \,, \tag{4.28}$$

over any closed surface $\Sigma^{(2)} \subset \mathcal{M}_T$, thus making $U_{2\pi}$ trivial on $\mathcal{M}_T$. As a consequence, $U_\beta$ satisfies $U_\beta = U_{\beta+2\pi}$ and implements a $U(1)^{(2)}$ symmetry in the physical theory. At the same time, the bulk operators $U_{2\pi n}$ with $n \in \mathbb{Z}$ can terminate on $\mathcal{M}_T$ because of the boundary conditions, and attach to the (non-topological) Wilson lines of the physical theory on $\mathcal{M}_P$.

As for the $V_\alpha$ operators, consider first the non-genuine bulk operators when $\alpha k \in 2\pi\mathbb{Z}$. Their 4-dimensional part can be opened on $\mathcal{M}_T$

$$V_{2\pi n/k}(\Pi^{(3)}, \Pi^{(3)}_{\mathcal{M}_T}, \Omega^{(4)}) = \exp\left( \frac{in}{k} \int_{\Pi^{(3)}} b^{(3)} - \frac{in}{4\pi} \int_{\Omega^{(4)}} a^{(2)} \wedge a^{(2)} - \frac{in}{4\pi} \int_{\Pi^{(3)}_{\mathcal{M}_T}} A^{(1)} \wedge dA^{(1)} \right) \,, \tag{4.29}$$

where $\Pi^{(3)}_{\mathcal{M}_T} \subset \mathcal{M}_T$ and $\partial\Omega^{(4)} = \Pi^{(3)} \cup \Pi^{(3)}_{\mathcal{M}_T}$. Indeed, only for $\alpha k = 2\pi n$ a correctly quantized Chern-Simons term can act as a junction with $\mathcal{M}_T$. In the above configuration the bulk operator is anchored to the boundary via a cylinder, similar to what was described in Section 2.3. For $\alpha \in 2\pi\mathbb{Z}$, the operator, when pushed completely to the boundary, is trivialized by the boundary conditions (4.22) on $b^{(3)}$.

When $\alpha k \notin 2\pi\mathbb{Z}$, we can define a gauge invariant operator by stacking a TQFT instead of the Chern-Simons term at the junction with $\mathcal{M}_T$. Since $dA^{(1)}$ has quantized integrals on $\mathcal{M}_T$, we can safely use the $\mathcal{T}^\alpha[dA^{(1)}]$ theory:

$$V_\alpha(\Pi^{(3)}, \Pi^{(3)}_{\mathcal{M}_T}, \Omega^{(4)}) = \exp\left(\frac{i\alpha}{2\pi}\int_{\Pi^{(3)}} b^{(3)} - \frac{i\alpha k}{8\pi^2}\int_{\Omega^{(4)}} a^{(2)}\wedge a^{(2)} + \mathcal{T}^{\alpha k}[dA^{(1)}](\Pi^{(3)}_{\mathcal{M}_T})\right),$$
(4.30)

where we have emphasized that the TQFT is defined on $\Pi^{(3)}_{\mathcal{M}_T} \subset \mathcal{M}_T$. This operator is well defined for any values of $\alpha \in \mathbb{R}$ and reproduces (4.10) when the symmetry TFT bulk is shrunk, using (4.26). Since $V_{2\pi}$ is trivial on the boundary, we also have that $V_\alpha = V_{\alpha+2\pi}$ after slab compactification, so that these operators generate a non-invertible $U(1)^{(1)}$ symmetry.

When considering cylinder operators, equation (4.20) is slightly modified to take into account the presence of a boundary. For example, if $V_\beta$ is a non-invertible operator (4.30), we get

$$\left\langle V_\alpha(\Pi_1^{(3)}, \Pi^{(3)}_{\mathcal{M}_{T_1}}, \Omega_1^{(4)}) V_\beta(\Pi_2^{(3)}, \Pi^{(3)}_{\mathcal{M}_{T_2}}, \Omega_2^{(4)})\right\rangle$$

$$= \left\langle V_\alpha(\widetilde{\Pi}_1^{(3)}, \widetilde{\Pi}^{(3)}_{\mathcal{M}_{T_1}}, \widetilde{\Omega}_1^{(4)}) V_\beta(\Pi_2^{(3)}, \Pi^{(3)}_{\mathcal{M}_{T_2}}, \Omega_2^{(4)}) U_{\frac{k\alpha\beta}{2\pi}}(\Omega_2^{(4)} \cap \widetilde{\Omega}_1^{(4)}) \exp\left(\frac{-i\alpha}{2\pi}\int_{\Pi^{(3)}_{\mathcal{M}_{T_2}}\cap\widetilde{\Pi}^{(3)}_{\mathcal{M}_{T_1}}} \Phi_2^{(1)}\right)\right\rangle,$$
(4.31)

where $\Phi_2^{(1)}$ is a dynamical field of the $\mathcal{T}^{\beta k}$ TQFT defining $V_\beta$. The operator $U_{\frac{k\alpha\beta}{2\pi}}$ above is defined on a surface that terminates on $\mathcal{M}_T$ through a junction, which is a topological line. Furthermore, we can shrink the topological line by noticing that it is possible to define a surface $\Sigma^{(2)}_{\mathcal{M}_{T_2}} \subset \Pi^{(3)}_{\mathcal{M}_{T_2}}$ satisfying $\partial\Sigma^{(2)}_{\mathcal{M}_{T_2}} = \Pi^{(3)}_{\mathcal{M}_{T_2}} \cap \widetilde{\Pi}^{(3)}_{\mathcal{M}_{T_1}}$, to obtain

$$\left\langle V_\alpha(\widetilde{\Pi}_1^{(3)}, \widetilde{\Pi}^{(3)}_{\mathcal{M}_{T_1}}, \widetilde{\Omega}_1^{(4)}) V_\beta(\Pi_2^{(3)}, \Pi^{(3)}_{\mathcal{M}_{T_2}}, \Omega_2^{(4)}) U_{\frac{k\alpha\beta}{2\pi}}\left((\Omega_2^{(4)} \cap \widetilde{\Omega}_1^{(4)}) \cup \Sigma^{(2)}_{\mathcal{M}_{T_2}}\right)\right\rangle.$$
(4.32)

A similar reasoning can be repeated for the invertible operator (4.29) and also leads to the above expression. Eventually, the $U_{\frac{k\alpha\beta}{2\pi}}$ operator generated at the intersection of two cylinders does not terminate on $\mathcal{M}_T$ but partially lies on it, as represented in the left hand side of Figure 2.

From the physical theory perspective, the relation (4.32) implies that when two operators $\mathcal{D}_\alpha$ and $\mathcal{D}_\beta$ intersect each other, they generate an open $U_{\frac{k\alpha\beta}{2\pi}}$ operator terminating on

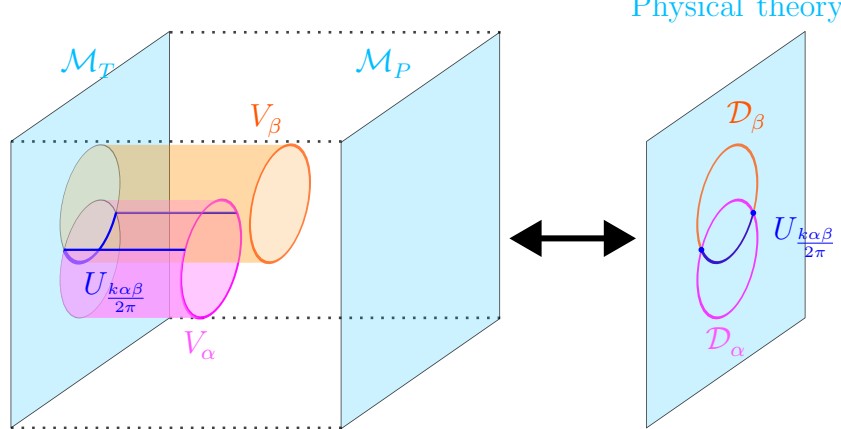

Figure 2: When two cylinder operators $V_\alpha$ and $V_\beta$ intersect each other, an open $U_{\frac{k\alpha\beta}{2\pi}}$ is generated. In the physical theory, this implies that open 2-form symmetry operators are generated at intersections of 1-form symmetry operators. This is a consequence of the non-trivial higher-group-like structure of the theory.

their intersection:

$$\left\langle \mathcal{D}_\alpha(\Pi_1^{(3)})\mathcal{D}_\beta(\Pi_2^{(3)}) \right\rangle = \left\langle \mathcal{D}_\alpha(\widetilde{\Pi}_1^{(3)})\mathcal{D}_\beta(\Pi_2^{(3)})U_{\frac{k\alpha\beta}{2\pi}}(\Sigma_\Pi^{(2)}) \right\rangle, \qquad (4.33)$$

where $\partial\Sigma_\Pi^{(2)} = \widetilde{\Pi}_1^{(3)} \cap \Pi_2^{(3)}$. The correspondence between (4.32) and (4.33) is represented in Figure 2. This relation follows from the non-trivial higher-group-like structure of the theory, which enables open 2-form symmetry operators to be generated at the intersections of 1-form symmetry operators. A key step in deriving these relations is the use of cylinder operators instead of genuine bulk operators. Only the intersections of non-genuine $V_\alpha$ operators have the correct dimensionality to reproduce (4.33).

Finally, let us consider which bulk operators terminate on the magnetic 't Hooft surfaces of the five-dimensional physical theory. We expect the operators $V_{2\pi m}$ to end on 't Hooft surfaces with charge $m \in \mathbb{Z}$. However, if we want them to end on the topological boundary, we must first define a genuine version of the bulk operators $V_{2\pi m}$. Since $a^{(2)}$ does not have quantized fluxes in the bulk, we cannot use the $\mathcal{T}^\alpha$ TQFT. Instead, we can use a very similar TQFT, defined as follows (see [19]):

$$\widehat{\mathcal{T}}^{2\pi mk}[a^{(2)}](\Pi^{(3)}) = \frac{i}{2\pi}\int_{\Pi^{(3)}} -\frac{mk}{2}C^{(1)} \wedge dC^{(1)} + \phi^{(1)}(dC^{(1)} + a^{(2)}), \qquad (4.34)$$

where $C^{(1)}$ and $\phi^{(1)}$ are respectively $U(1)$ and $\mathbb{R}$ gauge fields. Note that this TQFT requires

$mk \in \mathbb{Z}$. The genuine bulk operators are then

$$V_{2\pi m}(\Pi^{(3)}) = \exp\left( im \int_{\Pi^{(3)}} b^{(3)} + \widehat{\mathcal{T}}^{2\pi mk}[a^{(2)}](\Pi^{(3)}) \right) . \tag{4.35}$$

These operators are gauge invariant provided that $C^{(1)}$ and $\phi^{(1)}$ transform under bulk gauge transformations as

$$C^{(1)} \to C^{(1)} + d\lambda_C^{(0)} - \lambda_a^{(1)} , \qquad \phi^{(1)} \to \phi^{(1)} + d\lambda_\phi^{(0)} - mk\lambda_a^{(1)} . \tag{4.36}$$

In order to see that such operators do indeed become 't Hooft surfaces after slab compactification, we need to impose boundary conditions on both $\mathcal{M}_T$ and $\mathcal{M}_P$ also for the TQFT fields. Mimicking (2.50), we can impose

$$\int (\phi^{(1)} + mkA^{(1)}) \in 2\pi\mathbb{Z} , \qquad \int (C^{(1)} + A^{(1)}) \in 2\pi\mathbb{Z} , \tag{4.37}$$

through the following boundary action:

$$\frac{i}{2\pi} \int d\Psi^{(0)} \wedge (C^{(1)} + A^{(1)}) - \frac{mk}{2} A^{(1)} \wedge C^{(1)} . \tag{4.38}$$

However, this action can only define consistent boundary conditions if $mk \in 2\mathbb{Z}$. Indeed, the boundary constraints involve two edge modes, $\Psi^{(0)}$ and $\Phi^{(0)}$ satisfying

$$C^{(1)} + A^{(1)} = d\Phi^{(0)} , \qquad \phi^{(1)} + \frac{mk}{2}(A^{(1)} - C^{(1)}) = d\Psi^{(0)} . \tag{4.39}$$

Requiring the gauge invariance of these constraints imposes:

$$\Phi^{(0)} \to \Phi^{(0)} + 2\pi n_\Phi + \lambda_C^{(0)} + \lambda_A^{(0)} , \quad \Psi^{(0)} \to \Psi^{(0)} + 2\pi n_\Psi + \lambda_\phi^{(0)} + \frac{mk}{2}\left(\lambda_A^{(0)} - \lambda_C^{(0)}\right) . \tag{4.40}$$

Since $\lambda_A^{(0)}$ and $\lambda_C^{(0)}$ are both $U(1)$ 0-forms, the gauge transformation of $\Psi^{(0)}$ is actually well defined only if $mk \in 2\mathbb{Z}$. When $mk$ is odd, the TQFT (4.34) depends on the spin structure of $\Pi^{(3)}$ and has fermionic lines [41], and imposing topological boundary conditions is more subtle. Here for simplicity we are treating explicitly only the simpler case where $mk$ is even.

On $\mathcal{M}_P$ one further introduces the term

$$\frac{1}{4\pi L^2} \int C^{(1)} \wedge *C^{(1)} , \tag{4.41}$$

to finally get after slab compactification on $\Pi^{(3)}$

$$\int \frac{L^2}{4\pi} \left( d\Psi^{(0)} - \frac{mk}{2} A^{(1)} \right) \wedge * \left( d\Psi^{(0)} - \frac{mk}{2} A^{(1)} \right) + \frac{i}{2\pi} d\Psi^{(0)} \wedge A^{(1)} \,. \qquad (4.42)$$

Magnetic surfaces of the physical theory obtained after the slab compactification will be dressed by this QFT.[21] This dressing is necessary to define a gauge invariant operator when $k \neq 0$. Indeed, similarly to the definition of 't Hooft lines in the Goldstone Maxwell theory, (3.12), magnetic surfaces in the physical theory can be defined as the boundary of an open symmetry operator of the electric symmetry:

$$H_m(\Sigma^{(2)}) = \exp \left( -2\pi m \int_{\Pi^{(3)}_\Sigma} *dA^{(1)} + \frac{imk}{4\pi} \int_{\Pi^{(3)}_\Sigma} A^{(1)} \wedge dA^{(1)} \right) \,, \qquad (4.43)$$

with $\partial\Pi^{(3)}_\Sigma = \Sigma^{(2)}$. Under a gauge transformation of parameter $\lambda_A^{(0)}$, the magnetic surface defined with the formula above transforms as

$$H_m(\Sigma^{(2)}) \to H_m(\Sigma^{(2)}) \exp \left( -\frac{imk}{4\pi} \int_{\Sigma^{(2)}} d\lambda_A^{(0)} \wedge A^{(1)} \right) \,. \qquad (4.44)$$

This non-trivial transformation is exactly canceled by the transformation of the QFT (4.42).

## Acknowledgements

We thank Jeremias Aguilera Damia, Andrea Antinucci, Pietro Benetti Genolini, Francesco Benini, Daniel Brennan, Giovanni Galati, Giovanni Rizi and Elise Paznokas for helpful discussions. AA, RA and LT are respectively a Research Fellow, a Research Director and a Postdoctoral Researcher (on leave) of the F.R.S.-FNRS (Belgium). This research is further supported by IISN-Belgium (convention 4.4503.15) and through an ARC advanced project.

## A. Generalities on $U(1)_A^{(0)} \times U(1)_C^{(0)}$ 't Hooft Anomalies

In this work we follow for the anomalies the conventions of [4]. In particular we have that the anomaly functional $\mathcal{A}$ is captured by:

$$\mathcal{A} = 2\pi i \int \mathcal{I}^{(d)} \,, \quad d\mathcal{I}^{(d)} = \delta\mathcal{I}^{(d+1)} \,, \quad d\mathcal{I}^{(d+1)} = \mathcal{I}^{(d+2)} \,, \qquad (A.1)$$

---

[21]This QFT describes a compact boson, with $A^{(1)}$ acting as a background field for a linear combination of its momentum and winding symmetries.

where $\mathcal{I}^{(d+1)}$ and $\mathcal{I}^{(d+2)}$ are local expressions in the background fields while $\mathcal{I}^{(d)}$ also depends on background gauge parameters.

Let us consider a four-dimensional theory with global symmetries:

$$U(1)_{\mathsf{A}}^{(0)} \times U(1)_{\mathsf{C}}^{(0)} . \tag{A.2}$$

We will focus on 't Hooft anomalies that are completely encoded by the following 6-form anomaly polynomial:

$$\mathcal{I}^{(6)} = \frac{1}{(2\pi)^3} \left( \frac{k_{\mathsf{A}^3}}{6} \mathsf{F}_{\mathsf{A}}^{(2)} \wedge \mathsf{F}_{\mathsf{A}}^{(2)} \wedge \mathsf{F}_{\mathsf{A}}^{(2)} + \frac{k_{\mathsf{A}^2\mathsf{C}}}{2!} \mathsf{F}_{\mathsf{A}}^{(2)} \wedge \mathsf{F}_{\mathsf{A}}^{(2)} \wedge \mathsf{F}_{\mathsf{C}}^{(2)} \right.$$
$$\left. + \frac{k_{\mathsf{A}\mathsf{C}^2}}{2!} \mathsf{F}_{\mathsf{A}}^{(2)} \wedge \mathsf{F}_{\mathsf{C}}^{(2)} \wedge \mathsf{F}_{\mathsf{C}}^{(2)} + \frac{k_{\mathsf{C}^3}}{6} \mathsf{F}_{\mathsf{C}}^{(2)} \wedge \mathsf{F}_{\mathsf{C}}^{(2)} \wedge F_{\mathsf{C}}^{(2)} \right) . \tag{A.3}$$

The anomaly coefficients are given by:

$$k_{A^3,C^3} = \sum_i (q_{A,C}^i)^3 , \quad k_{\mathsf{A}^2\mathsf{C}} = \sum_i (q_A^i)^2 q_C^i , \quad k_{\mathsf{A}\mathsf{C}^2} = \sum_i q_A^i (q_C^i)^2 , \tag{A.4}$$

where the sum runs over free fermions carrying the above charges. These coefficients obey the following quantization conditions:

$$k_{\mathsf{A}^3} = 6l_A + n_A , \quad k_{\mathsf{C}^3} = 6l_C + n_C , \quad k_{\mathsf{A}^2\mathsf{C}} = 2p + r , \quad k_{\mathsf{A}\mathsf{C}^2} = 2\widetilde{p} + r , \tag{A.5}$$

where $l_{A,C}, n_{A,C}, p, \widetilde{p}, r \in \mathbb{Z}$. Note that $r$ appears in both $k_{\mathsf{A}^2\mathsf{C}}$ and $k_{\mathsf{A}\mathsf{C}^2}$, in a theory leading to a 2-group this is important since requiring that $k_{\mathsf{A}\mathsf{C}^2} = 0$ implies that $k_{\mathsf{A}^2\mathsf{C}} \in 2\mathbb{Z}$.

As we are interested in gauging the symmetry $U(1)_{\mathsf{C}}^{(0)}$ we need to ensure that $d * j_{\mathsf{C}}^{(1)} = 0$ which is identical to require that $\mathcal{A}_{\mathsf{C}} = 0$. It turns out that this completely fixes the anomaly of the $U(1)_{\mathsf{A}}^{(0)}$ symmetry to be:

$$\mathcal{A}_{\mathsf{A}} = \frac{i}{4\pi^2} \int_{\mathcal{M}_4} \lambda_{\mathsf{A}}^{(0)} \left( \frac{k_{\mathsf{A}^3}}{6} \mathsf{F}_{\mathsf{A}}^{(2)} \wedge \mathsf{F}_{\mathsf{A}}^{(2)} + \frac{k_{\mathsf{A}^2\mathsf{C}}}{2} \mathsf{F}_{\mathsf{A}}^{(2)} \wedge \mathsf{F}_{\mathsf{C}}^{(2)} + \frac{k_{\mathsf{A}\mathsf{C}^2}}{2} \mathsf{F}_{\mathsf{C}}^{(2)} \wedge \mathsf{F}_{\mathsf{C}}^{(2)} \right) . \tag{A.6}$$

The above anomaly functional implies the non-conservation equation appearing in (2.3).

## B. Linking Invariants

In this work we follow the conventions on linking invariants from [61]. Let us consider a set of $N$ manifolds $M_i^{(p_i)}$, $i = 1, 2, ..., N$, of dimensions $p_i$ in a spacetime that for simplicity we take to be $S^{(d)}$. For $N = 2$, the 2-component link between $M_1^{(p_1)}$ and $M_2^{(p_2)}$ is defined as

$$\text{Link}(M_1^{(p_1)}, M_2^{(p_2)}) := \int_{S^{(d)}} \delta(\widehat{M}_1^{(p_1+1)}) d\delta(\widehat{M}_2^{(p_2+1)}), \tag{B.1}$$

where $\widehat{M}_i^{(p_i+1)}$ is a $(p_i + 1)$-dimensional manifold satisfying $\partial \widehat{M}_i^{p_i+1} = M_i^{p_i}$ and $\delta(\widehat{M}_i^{(p_i+1)})$ denotes its Poincaré dual. Note that the above integral is non-trivial if and only if the integrand is a $d$-form requiring:

$$p_1 + p_2 + 1 = d. \tag{B.2}$$

The generalization to $N > 2$ of the above story is the $N$-component linking of type $k$ defined as:

$$\text{Link}(M_1^{(p_1)}, ..., M_N^{(p_N)})_k := \int_{S^{(d)}} \delta(\widehat{M}_1^{(p_1+1)}) \ldots \delta(\widehat{M}_{N-k}^{(p_{N-k}+1)}) d\delta(\widehat{M}_{N-k+1}^{(p_{N-k+1}+1)}) \ldots d\delta(\widehat{M}_N^{(p_N+1)}). \tag{B.3}$$

As before, the integrand must be a $d$-form, therefore:

$$\sum_{i=1}^{N-k} (p_i + 1) + \sum_{j=N-k+1}^{N} p_j = (N-1)d. \tag{B.4}$$

## C. Mutual Action of Non-invertible V and T

The genuine operators $T_m$ and $V_\alpha$, defined in (2.39) and (2.46) can be written as follow:

$$V_\alpha(\Pi^{(3)}) = V_\alpha(\Pi^{(3)}, \Omega^{(4)}) \mathbf{V}_\alpha(\Pi^{(3)}, \Omega^{(4)}), \quad T_m(\Sigma^{(2)}) = T_m(\Sigma^{(2)}, \Omega^{(3)}) \mathbf{T}_m(\Sigma^{(2)}, \Omega^{(3)}), \tag{C.1}$$

where $V_\alpha(\Pi^{(3)}, \Omega^{(4)})$ and $T_m(\Sigma^{(2)}, \Omega^{(3)})$ are the non-genuine operators defined in (2.19) and (2.20). The non-genuine operators $\mathbf{V}_\alpha(\Pi^{(3)}, \Omega^{(4)})$ and $\mathbf{T}_m(\Sigma^{(2)}, \Omega^{(3)})$ are the TQFTs used in (2.46) and (2.39) respectively, with appropriate anomaly inflow actions:

$$\begin{aligned}
\mathbf{V}_\alpha(\Pi^{(3)}, \Omega^{(4)}) &= \exp\left( \mathcal{T}^\alpha(\Pi^{(3)}) + \frac{i\alpha}{8\pi^2} \int_{\Omega^{(4)}} f^{(2)} \wedge f^{(2)} \right), \\
\mathbf{T}_m(\Sigma^{(2)}, \Omega^{(3)}) &= \exp\left( \mathcal{T}_{2d}^{mk_{\mathsf{AC}^2}}(\Sigma^{(2)}) + \frac{imk_{\mathsf{AC}^2}}{2\pi} \int_{\Omega^{(3)}} A^{(1)} \wedge f^{(2)} \right).
\end{aligned} \tag{C.2}$$

Note that the anomaly inflow action of the TQFT exactly cancel the non-genuine part of the non-genuine $V$ and $T$. According to relation (C.1), the mutual action of genuine $V$ and $T$ can be obtained from the mutual action of the 4 non-genuine operators appearing in the right hand side. The mutual action of non-genuine $V$ and $T$ is given by (2.24). The mutual action of $\mathbf{T}$ and $\mathbf{V}$ is trivial since the operators involved in the gauging defining the first TQFT are not charged under the symmetries gauged to define the second one. We now consider the action of non-genuine $V$ and $T$ on $\mathbf{T}$ and $\mathbf{V}$. These are respectively given by:

$$\left\langle V_\alpha(\Pi^{(3)}, \Omega^{(4)}) \mathbf{T}_m(\Sigma^{(2)}, \Omega^{(3)}) \right\rangle = \left\langle V_\alpha(\widetilde{\Pi}^{(3)}, \widetilde{\Omega}^{(4)}) \mathbf{T}_m(\Sigma^{(2)}, \Omega^{(3)}) \times \right.$$
$$\left. U_{-\alpha m k_{\mathsf{AC}^2}}(\widetilde{\Omega}^{(4)} \cap \Omega^{(3)}) \exp\left( \frac{i\alpha}{2\pi} \int_{\Sigma^{(2)} \cap \widetilde{\Omega}^{(4)}} \phi^{(1)} \right) \right\rangle, \tag{C.3}$$

$$\left\langle T_m(\Sigma^{(2)}, \Omega^{(3)}) \mathbf{V}_\alpha(\Pi^{(3)}, \Omega^{(4)}) \right\rangle = \left\langle T_m(\Sigma^{(2)}, \Omega^{(3)}) \mathbf{V}_\alpha(\Pi^{(3)}, \Omega^{(4)}) \times \right.$$
$$\left. U_{-\alpha m k_{\mathsf{AC}^2}}(\widetilde{\Omega}^{(4)} \cap \Omega^{(3)}) \exp\left( -im \int_{\Omega^{(3)} \cap \widetilde{\Pi}^{(3)}} \Phi^{(1)} \right) \right\rangle, \tag{C.4}$$

The action of the non-genuine operators $V_\alpha$ and $T_m$ on the anomaly inflow actions of the TQFTs is responsible for the presence $U_{-\alpha m k_{\mathsf{AC}^2}}$ on the right hand side of these relations. These operator are the orientation reversal of the surface operator generated in (2.24). This is due to the fact that the anomaly inflow actions of the TQFTs are the orientation reversal of the non-genuine part of (2.19) and (2.20). From relations and (2.24), (C.3) and (C.4), we directly obtain the following relation:

$$\left\langle V_\alpha(\Pi^{(3)}) T_m(\Sigma^{(2)}) \right\rangle = \left\langle V_\alpha(\widetilde{\Pi}^{(3)}) T_m(\Sigma^{(2)}) U_{-m\alpha k_{\mathsf{AC}^2}}(\widetilde{\Omega}^{(4)} \cap \Omega^{(3)}) \times \right.$$
$$\left. \exp\left( \frac{i\alpha}{2\pi} \int_{\Sigma^{(2)} \cap \widetilde{\Omega}^{(4)}} \phi^{(1)} \right) \exp\left( -im \int_{\Omega^{(3)} \cap \widetilde{\Pi}^{(3)}} \Phi^{(1)} \right) \right\rangle. \tag{C.5}$$

The genuine operators $V_\alpha$ and $T_m$ satisfy a similar relation as their non-genuine counterpart in (2.24). There are, however, two differences between (2.24) and (C.5). First, the two $U_{m\alpha k_{\mathsf{AC}^2}}$ operators appearing on the right hand side of both relations have opposite orientations. Indeed, we get two $U_{-m\alpha k_{\mathsf{AC}^2}}$ operators from relations (C.3) and (C.4). One of them cancels the operator appearing in (2.24) while the other one remains in (C.5). The second difference is that the open $U_{m\alpha k_{\mathsf{AC}^2}}$ terminates topologically on $V_\alpha$ and $T_m$ through

line operators of their respective TQFT. The presence of these topological junction operators is needed to guarantee that the left hand side of (C.5) is independent of the choice of $\widetilde{\Omega}^{(4)}$ and $\Omega^{(3)}$. From the physical theory perspective, relation (C.5) implies that when a non-invertible operator crosses a 't Hooft line, the 't Hooft line gets attached to an open magnetic symmetry operator.

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
