# Peer review of "U(1) Gauging, Continuous TQFTs, and Higher Symmetry Structures"

_SciPost Physics_

## Round 2 · Referee Report · Anonymous (Referee 1) · 2025-5-1

Report

The paper investigates the symmetry structures arising in quantum field theories (QFTs) with $U(1)^{(0)}_{\mathsf{A}} \times U(1)^{(0)}_{\mathsf{C}}$ global symmetries characterized by specific 't Hooft anomalies ($k_{\mathsf{A}^3}, k_{\mathsf{A}^2\mathsf{C}}, k_{\mathsf{A}\mathsf{C}^2}, k_{\mathsf{C}^3}$). It focuses specifically on the consequences of gauging the $U(1)^{(0)}_{\mathsf{C}}$ symmetry (assuming $k_{\mathsf{C}^3}=0$), particularly in scenarios where this leads to the simultaneous emergence of higher-group symmetries (mixing 0-form and 1-form symmetries) and non-invertible symmetries. The authors term this combination the generalized Abelian symmetry structure and provide a unified framework for its analysis using the Symmetry Topological Field Theory (SymTFT).

The work details the 5d SymTFT action governing this generalized structure, where the $U(1)^{(0)}_{\mathsf{A}}$ symmetry becomes non-invertible (due to $k_{\mathsf{A}\mathsf{C}^2} \neq 0$) while exhibiting 2-group mixing with the emergent magnetic $U(1)^{(1)}_{\mathsf{C}}$ symmetry (due to $k_{\mathsf{A}^2\mathsf{C}} \neq 0$). It clarifies the behavior of the corresponding bulk SymTFT operators $V_\alpha$ and $T_m$: $V_\alpha$ (related to $U(1)^{(0)}_{\mathsf{A}}$) becomes non-invertible and may require non-genuine descriptions like cylinder attachments, while $T_m$ (related to $U(1)^{(1)}_{\mathsf{C}}$) can often be rendered genuine via stacking with appropriate 2d TQFTs incorporating both anomaly coefficients. Furthermore, the paper analyzes the scheme dependence of anomaly coefficients ($k_{\mathsf{A}^3}, k_{\mathsf{A}^2\mathsf{C}}$) under field redefinitions within the SymTFT, demonstrating how non-invertibility ($k_{\mathsf{A}\mathsf{C}^2} \neq 0$) influences these shifts. The SymTFT framework is also shown to naturally explain the necessity of quantum mechanics dressings for 't Hooft lines in certain phases (e.g., Goldstone-Maxwell) when $k_{\mathsf{A}\mathsf{C}^2} \neq 0$.

The paper is very well-written and clearly presented. It merits publication in SciPost once the following minor points are addressed.

Requested changes

  1. In Section 2.4, the authors point out that the presence of a non-invertible symmetry can induce a 2-group coefficient. It would be helpful if the authors could comment on whether this statement also applies to finite discrete symmetries.

  2. Minor typos: 2.1 Above Eq. (2.10): as follow: -> as follows: 2.2 Above Eq. (2.61): whose associated edge modes gauge transformation are -> whose associated edge mode gauge transformations are. 2.3 Below Eq. (C.4): These operator are -> These operators are

Recommendation

Ask for minor revision

  • validity: high
  • significance: high
  • originality: high
  • clarity: high
  • formatting: excellent
  • grammar: excellent

Author:  Riccardo Argurio  on 2025-06-24  [id 5595]

(in reply to Report 1 on 2025-05-01)

We would like to thank the referee for the careful reading of our manuscript. We have fixed all the typos that were spotted.
Concerning the first point, we think that the referee has raised a very interesting question, that certainly deserves further scrutiny. We however decided not to mention in the revised version the relation between non-invertible symmetries and higher-groups in the discrete case, since we feel that it would bring us too far from the main focus of our paper, which is on continuous symmetries. Indeed, we believe that though there is probably an interesting interplay, its details are presumably very different in the discrete case. Note for instance that the simple shifts (2.65) and (2.67), refer inherently to quantities that can be defined only in the continuous case.

---

## Round 2 · Referee Report · Anonymous (Referee 2) · 2025-5-18

Report

This paper studies the categorical symmetry structures that arise on a theory with global symmetry $U(1)_a\times U(1)_b$ upon gauging one of the factors. In general the two factors can have mixed anomalies, which can give rise to 2-groups, non-invertible symmetries, or (generically) a combination of both.

The authors clarify what happens in this last case, using the SymTFT description of the symmetries, a bulk (that is, one higher dimensional) theory that encodes the symmetry information.

The analysis in this paper is very interesting, and the paper is very well written. This is also a very timely topic. I will be happy to recommend the paper for publication once the following two minor points are addressed:

  • One of the guiding principles the authors use for much of the analysis, when it comes to constructing the theory on the gapped boundary, is that of gauge invariance. I think it would benefit the paper if the authors elaborated on why they impose this requirement, and when exactly does this need to hold. As far as I can tell it makes sense to consider boundary actions which are not gauge invariant, see for example (A.15) in https://arxiv.org/abs/hep-th/0108152 . And in some parts of the analysis, for example in the paragraph below (3.4) in the paper (pg.24), the authors instead seem to impose gauge invariance by restricting the set of gauge transformations they consider. But this is something that one can always do for any action one can think of, gauge invariant or not. So I think it would help the reader if the rules of the game were explicitly laid out from the beginning. (To be clear: I have no issue with the actions the authors actually write, but the motivation for invoking gauge invariance when deriving them is not transparent to me.)

  • At the bottom of pg. 33 it is stated that for $Y^{(6)}$ spin we can take $k\in\mathbb{Z}$. I don't think this is true, for this to hold one needs to add a term

    $$2\pi \frac{k}{24(2\pi)^3}\int_{Y^{(6)}} F^{(2)}\wedge p_1(TY^{6})$$
    to (4.2) (or its associated Chern-Simons form to (4.1)), so that the combined integrand is the index density for a Weyl fermion charged under $U(1)$ in six dimensions. (For a concrete counterexample to the claim in the paper, take $Y^{(6)}$ and $Z^{(6)}$, with $\partial Y^{(6)}=\partial Z^{(6)} = M^{(5)}$, such that $Y^{(6)}-Z^{(6)}=\mathbb{C}\mathbb{P}^3$, and take $F^{(2)}$ the hyperplane class $\ell\in H^2(\mathbb{C}\mathbb{P}^3;\mathbb{R})$, so that $\int_{Y^{(6)}-Z^{(6)}}(F^{(2)})^3=1$.)

Recommendation

Ask for minor revision

---

## Editorial Decision

resubmitted